# Haploid genetic screens identify SPRING/C12ORF49 as a determinant of SREBP signaling and cholesterol metabolism

Anke Loregger[1,11], Matthijs Raaben[2,11], Joppe Nieuwenhuis[2,11], Josephine M.E. Tan[1], Lucas T. Jae[2,3], Lisa G. van den Hengel[2], Sebastian Hendrix[1], Marlene van den Berg[1], Saskia Scheij[1], Ji-Ying Song[4], Ivo J. Huijbers[5], Lona J. Kroese[5], Roelof Ottenhoff[1], Michel van Weeghel[6], Bart van de Sluis[7,8], Thijn Brummelkamp[2,9,10,12✉] & Noam Zelcer[1,12✉]

The sterol-regulatory element binding proteins (SREBP) are central transcriptional regulators of lipid metabolism. Using haploid genetic screens we identify the SREBP Regulating Gene (*SPRING/C12ORF49*) as a determinant of the SREBP pathway. SPRING is a glycosylated Golgi-resident membrane protein and its ablation in Hap1 cells, Hepa1-6 hepatoma cells, and primary murine hepatocytes reduces SREBP signaling. In mice, *Spring* deletion is embryonic lethal yet silencing of hepatic *Spring* expression also attenuates the SREBP response. Mechanistically, attenuated SREBP signaling in SPRING^KO cells results from reduced SREBP cleavage-activating protein (SCAP) and its mislocalization to the Golgi irrespective of the cellular sterol status. Consistent with limited functional SCAP in SPRING^KO cells, reintroducing SCAP restores SREBP-dependent signaling and function. Moreover, in line with the role of SREBP in tumor growth, a wide range of tumor cell lines display dependency on *SPRING* expression. In conclusion, we identify SPRING as a previously unrecognized modulator of SREBP signaling.

[1] Department of Medical Biochemistry, Amsterdam UMC, Amsterdam Cardiovascular Sciences and Gastroenterology and Metabolism, University of Amsterdam, Meibergdreef 9, 1105AZ Amsterdam, The Netherlands. [2] Oncode Institute, Division of Biochemistry, The Netherlands Cancer Institute, Plesmanlaan 121, 1066CX Amsterdam, The Netherlands. [3] Gene Center and Department of Biochemistry, Ludwig-Maximilians-Universität München, Feodor Lynen-Str. 25, 81377 Munich, Germany. [4] Division of Experimental Animal Pathology, The Netherlands Cancer Institute, Plesmanlaan 121, 1066CX Amsterdam, The Netherlands. [5] Mouse Clinic for Cancer and Aging (MCCA) Transgenic Facility, The Netherlands Cancer Institute, Plesmanlaan 121, 1066CX Amsterdam, The Netherlands. [6] Laboratory of Genetic and Metabolic Diseases and Core Facility Metabolomics, Academic Medical Center of the University of Amsterdam, Meibergdreef 9, 1105AZ Amsterdam, The Netherlands. [7] Department of Pediatrics, University Medical Center Groningen, Antonius Deusinglaan 1, 9713AV Groningen, The Netherlands. [8] iPSC/CRISPR Center Groningen, University Medical Center Groningen, Antonius Deusinglaan 1, 9713AV Groningen, The Netherlands. [9] CeMM Research Center for Molecular Medicine of the Austrian Academy of Sciences, Lazarettgasse 14, A-1090 Vienna, Austria. [10] Cancer Genomics Center, Amsterdam, The Netherlands. [11]These authors contributed equally: Anke Loregger, Matthijs Raaben, Joppe Nieuwenhuis. [12]These authors jointly supervised this work: Thijn Brummelkamp, Noam Zelcer. ✉email: t.brummelkamp@nki.nl; n.zelcer@amsterdamumc.nl

Cellular sterol and fatty acid levels must be tightly controlled to ensure that these meet metabolic and growth demands[1]. Accordingly, loss of lipid homeostasis is associated with a wide range of human conditions, including cancer, neurodegeneration, and cardiovascular disease. The sterol-regulatory element binding proteins (SREBPs) are a family of transcription factors that control all facets of lipid metabolism by regulating the expression of a panel of genes that contain a sterol regulatory element (SRE) in their respective promoter regions[2–4]. There are three SREBP isoforms, SREBP1a, SREBP1c, and SREBP2[4–6], which are structurally similar, but activate a distinct set of genes and exhibit a divergent tissue distribution in vivo. SREBP1c primarily regulates genes implicated in fatty acid synthesis such as fatty acid synthase (FASN) and acetyl-CoA carboxylase (ACC)[5]. In contrast, SREBP2 is primarily implicated in the regulation of genes linked to cholesterol synthesis and uptake, including those encoding for the rate-limiting enzymes in cholesterol biosynthesis, 3-hydroxy-3-methylglutaryl-coenzyme A reductase (HMGCR) and squalene epoxidase (SQLE), and the low-density lipoprotein receptor (LDLR)[6]. The third isoform, SREBP1a, regulates genes involved in both sterol and fatty acid metabolism[4].

SREBPs are produced in their precursor form as membrane-associated endoplasmic reticulum (ER)-resident proteins that contain an N-terminal basic helix-loop-helix leucine zipper domain and a regulatory carboxyterminal region[5]. Under conditions of sufficient cellular sterols, SREBPs are retained in the ER through formation of a tripartite complex with the sterol-sensing SREBP cleavage-activating protein (SCAP)[7,8], and the ER anchor proteins insulin-induced genes 1 and 2 (INSIG1 and 2)[9,10]. A drop in cholesterol in the ER membrane leads to a conformational change in SCAP[11,12], which leads to dissociation from INSIG and facilitates COPII-mediated trafficking of the SCAP-SREBP complex to the Golgi[13–15]. In the Golgi, SREBP is proteolytically activated through sequential cleavage by the proteases S1P and S2P[16,17] (encoded by MBTPS1 and MBTPS2), which release the transcriptionally active N-terminal domain of SREBP. Translocation of this domain to the nucleus leads to transcriptional activation of target genes containing an SRE. As SREBPs contain an SRE in their promoter region, they induce their own expression in a positive feedback mechanism[4,18].

The final step in the SCAP-SREBP cycle is the less-well understood COPI-mediated retrograde transport of SCAP from the Golgi to the ER, which allows SCAP to reiteratively associate with newly synthesized SREBPs and INSIGs in the ER[19]. To identify unknown determinants of the SREBP pathway we applied a set of mammalian haploid genetic screens and report here the identification of the SREBP Regulating Gene SPRING (C12ORF49), as a previously unrecognized factor that governs SREBP activity in mammalian cells and in vivo in mice by controlling the level of functional SCAP.

## Results

**Haploid genetic screens link SPRING to the SREBP pathway**. The SREBP pathway is subject to exquisite regulation by a core set of molecular factors that include INSIGs, SCAP, MBTPS1, and MBTPS2[2]. To identify unknown SREBP regulators we developed two SREBP-focused mammalian haploid genetic screens. Using this approach, we interrogated SREBP signaling in an unbiased manner reasoning that any unknown regulator should be found in independent screens.

In the first screen, we evaluated the cholesterol-dependent regulation of SQLE, a rate-limiting enzyme in cholesterol biosynthesis and a bona fide transcriptional target of SREBP2[20,21]. To monitor the level of SQLE protein in live cells as a proxy for SREBP activity

we engineered Hap1 cells in which we introduced mNeon into the last coding exon of the endogenous SQLE locus using CRISPR/Cas9-based microhomology-mediated end-joining integration (Fig. 1a and Supplementary Fig. 1A). Consistent with SREBP-dependent regulation of SQLE expression, the level of the SQLE-mNeon fusion protein was low in Hap1-SQLE-mNeon cells when cultured in sterol-containing medium yet was markedly increased upon sterol-depletion (Fig. 1b, c). Moreover, similar to untagged SQLE, the levels of chimeric SQLE-mNeon protein were subject to cholesterol-stimulated proteasomal degradation (Supplementary Fig. 1B). Using this cell line, we screened for positive genetic regulators that are required for SREBP signaling as well as for negative determinants essential for cholesterol-mediated degradation of SQLE[22,23], as illustrated in Fig. 1d. Briefly, following mutagenesis, Hap1-SQLE-mNeon cells were first sterol-depleted and subsequently treated with water-soluble cholesterol to induce SQLE-mNeon degradation. Mutants with the 5% lowest and highest mNeon signal were isolated by FACS and the integration sites retrieved from genomic DNA and mapped, as previously reported[24]. Validating our screening approach, we identified strong enrichment of gene-trap insertions in the SQLE locus in the mNeon[LOW] population, alongside a similar enrichment in the established positive regulators of the SREBP pathway SCAP, MBTPS1, MBTPS2, and SREBF2 itself (Fig. 1e). Conversely, the strongest negative regulator of SQLE-mNeon found in our screen was the E3 ubiquitin ligase MARCH6, and its cognate E2 partner UBE2J2, which we have recently reported to be critical determinants of cholesterol-dependent degradation of SQLE[25–27]. Additionally, as expected our screen also identified INSIG1, an established negative regulator of the SREBP pathway. As such, this screen faithfully reports on cholesterol-dependent regulation of SQLE by the SREBP pathway. Amongst the known core SREBP activating genes, identified as positive regulators of SQLE expression, we also found an uncharacterized gene, C12ORF49, which is further referred to as SREBP-Regulating Gene (SPRING).

In a parallel screen, we leveraged our recent finding that Hap1 cells tolerate loss of the key de novo fatty acid synthesis enzyme, FASN, which is a canonical SREBP1-regulated gene[24]. We reasoned that in the absence of FASN and fatty acid synthesis Hap1 cells must rely on alternative survival pathways for acquiring fatty acids and growth. To test this idea, we generated independent Hap1-FASN[KO] cells (Supplementary Fig. 2A) and conducted a synthetic lethality screen, as previously reported[28]. Briefly, Hap1-FASN[KO] cells were mutagenized and expanded in culture for 12 days to allow depletion of lethal mutations. Synthetic genetic interactions were thereafter analyzed by comparing the results obtained in Hap1-FASN[KO] cells and WT Hap1 cells treated in the same manner. A total of 72 genes showed a synthetic genetic interaction in Hap1-FASN[KO] cells (Fig. 1f). Amongst these, a prominent SREBP signature encompassing the gene encoding SREBP2 itself, SREBF2, the genes encoding the core SREBP-activation machinery (SCAP, MBTPS1, and MBTPS2) and the uncharacterized gene SPRING were apparent in Hap1-FASN[KO] cells. Notably, these genes appeared to be non-essential in WT Hap1 cells. Additionally, a set of SREBP target genes including LDLR, ACSL1, ACSL3, and FABP5, that are implicated in lipid uptake, trafficking, and metabolism also displayed a synthetic interaction in Hap1-FASN[KO] cells (Supplementary Fig. 3A). These observations suggest that in the absence of FASN, Hap1 cells depend on SREBP-dependent lipid uptake and further support the idea that SPRING may be implicated in SREBP signaling.

Previously, we have reported a haploid genetic screen that revealed that the entry of Andes viruses into target cells critically depends on SREBP signaling and on the presence of cholesterol in the host-cell membrane[29]. When integrating the three SREBP-focused haploid screens we identify a core set of 5 shared genes

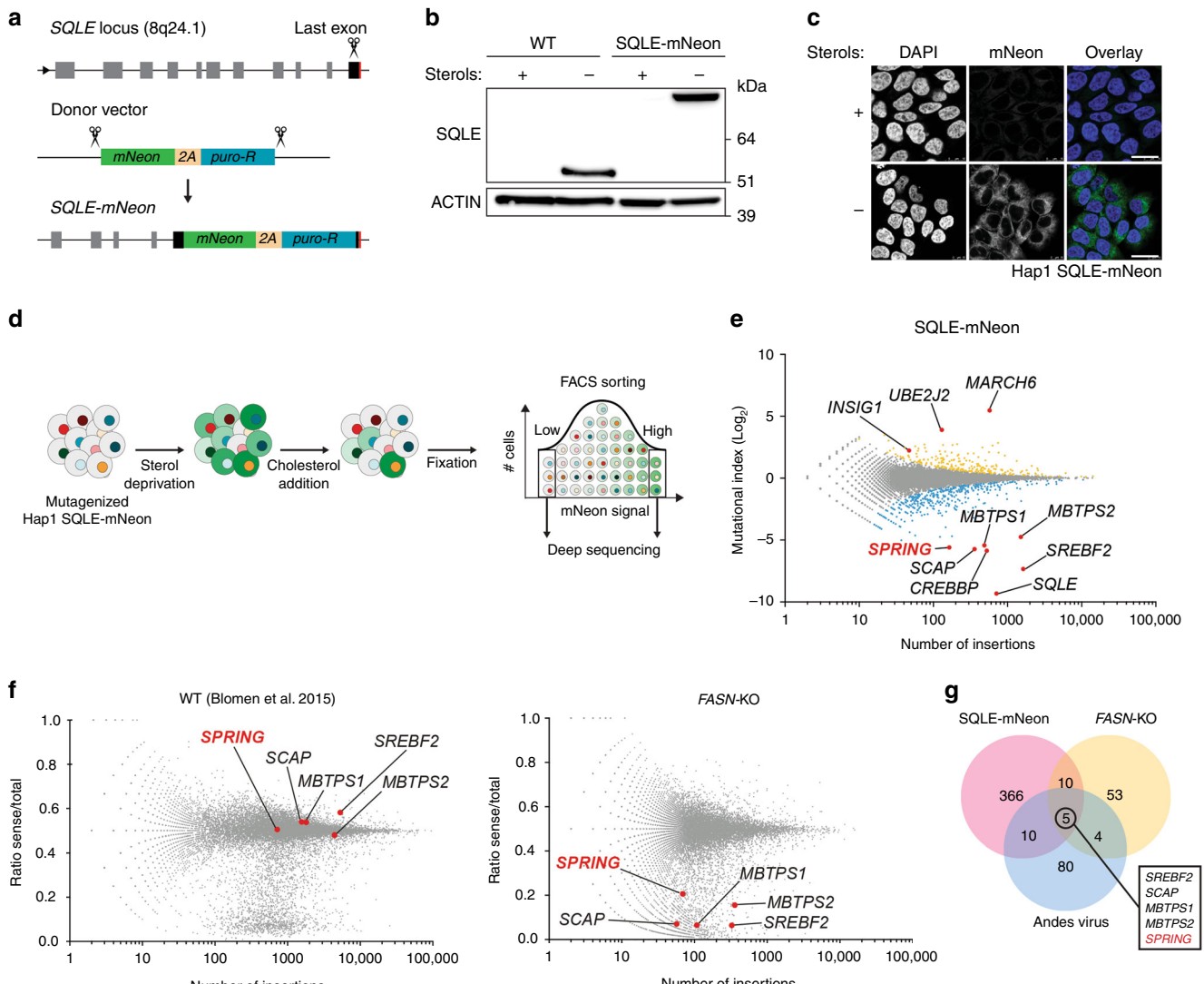

**Fig. 1 Haploid mammalian genetic screens identify SPRING as an SREBP regulator. a** Schematic illustration of CRISPR/Cas9-mediated targeting of the endogenous SQLE locus for in-frame integration of mNeon-2A-PURO. **b** Hap1 control and Hap1 SQLE-mNeon cells were cultured in the presence or absence of sterols and total cell lysates were immunoblotted as indicated. **c** Hap1 SQLE-mNeon cells were grown as in **b** on coverslips and subsequently fixed and counterstained with DAPI (nucleus). Cells were imaged and representative images are shown; scale bar, 25 μm. **d** Schematic depiction of SQLE-mNeon screen: a library of mutant Hap1 SQLE-mNeon cells was cultured in sterol-depletion medium for 24 h to induce SQLE expression. During the last 6 h β-methyl-cyclodextrin-cholesterol was added to initiate sterol-stimulated degradation of SQLE after which mNeon[HIGH] and mNeon[LOW] cells were isolated by FACS. **e** The log mutational index-scores (see Methods section) were plotted against the number of trapped alleles per gene. Statistically significant hits ($p < 0.05$) in the isolated mNeon[HIGH] and mNeon[LOW] cell populations are indicated in yellow and blue, respectively. **f** Comparison of gene-essentiality screens between control Hap1 cells and Hap1-FASN[KO] cells. Per gene, the ratio of sense/total orientation gene-trap insertions (y-axis) and the total number of insertions in a particular gene (x-axis) are plotted. FASN synthetic lethality screens show that SCAP, SREBF2, MBTPS1/2, and SPRING are significantly depleted in FASN[KO] cells. **g** Venn diagram indicating the number of unique and commonly identified genes in 3 individual SREBP-focused screens. **b**, **c** Representative images of three independent experiments are shown.

necessary for SREBP signaling that next to SREBF2, SCAP, MBTPS1, MTBPS2 also includes SPRING (Fig. 1g). In aggregate, our three independent screens strongly point towards SPRING acting as a positive regulator of the SREBP pathway to govern lipid metabolism.

**SPRING is a determinant of the SREBP pathway in cells.** As SPRING was identified in three independent screens in conjunction with the core SREBP activation machinery we reasoned that it could play a role in SREBP-mediated transcriptional control of lipid metabolism. To test this, we engineered Hap1 cells lacking SPRING and tested their response to sterol-depletion (Fig. 2a, b). Cells lacking SPRING had reduced protein levels of the SREBP

targets SQLE, SQS, and INSIG1 under basal culture conditions, and were unable to increase abundance of these under sterol-depletion condition. Similarly, in contrast to control Hap1 cells which, as expected, robustly increased the mRNA expression of the SREBP-transcriptional targets HMGCR, LDLR, SQLE, and FASN in response to sterol-depletion, in Hap1-SPRING[KO] cells basal expression of these genes was reduced and the response to sterol-depletion was largely abrogated. Notably, mRNA and protein levels of SPRING were not sensitive to the cellular sterol status. As a functional consequence, Hap1-SPRING[KO] cells exhibited markedly reduced levels of surface LDLR protein and failed to increase LDL uptake under sterol-deprived conditions (Fig. 2c, d and Supplemental Fig. 2E). Importantly, regulation of the SREBP

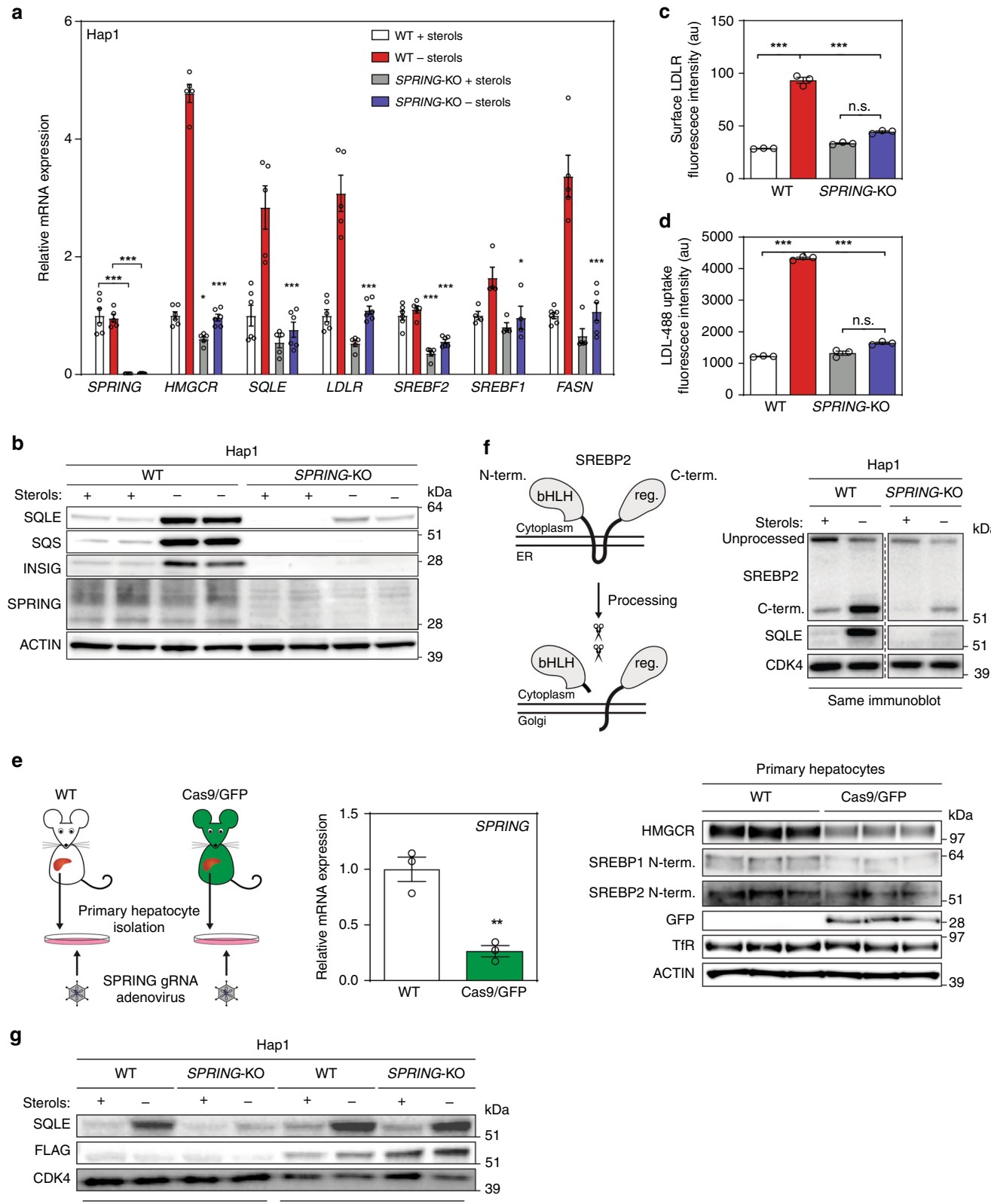

pathway was not restricted to Hap1 cells but was also observed in hepatocytes, which represent a physiological model for SREBP activity. Similar to our finding in Hap1 cells, loss of *Spring* in murine Hepa1-6 hepatoma cells and in primary mouse hepatocytes attenuated activity of the SREBP pathway (Fig. 2e and Supplementary Fig. 4).

Proteolytic processing of SREBPs is a prerequisite for their function as transcription factors, raising the possibility that SPRING may govern SREBPs at the level of their processing (Fig. 2f). Notably, we found that proteolytic processing of SREBP2 can occur in the absence of *SPRING*. However, we observed that absence of *Spring* profoundly reduced the precursor

**Fig. 2 Ablation of SPRING reduces SREBP signaling in cell lines and primary mouse hepatocytes. a, b** Hap1-WT and Hap1-SPRING[KO] cells were cultured in the presence or absence of sterols. Subsequently, cells were harvested for **a** gene expression analysis by qPCR as indicated ($N = 5$ biologically independent samples), and **b** immunoblotting as indicated ($N = 3$). **c, d** Cells were grown as in **a, b** and were (**c**) stained with an anti-LDLR-APC antibody, or **d** incubated with 5 μg/ml DyLight 488-labeled LDL for 1 h. Subsequently, cells were fixed and analyzed by FACS ($N = 3$ biologically independent samples). **e** Primary mouse hepatocytes were isolated from C57BL/6J WT and Cas9 knock-in mice and 4 h post isolation infected with Ad-3x -sgRNA-Spring adenoviral particles. Cells were sterol-depleted for 16 h and harvested for immunoblotting and gene expression analysis as indicated ($N = 3$ biologically independent samples). **f** Schematic representation of SREBP processing (left). Total cell lysates from Hap1-WT and Hap1-SPRING[KO] cells were immunoblotted as indicated (right). A representative image of at least three independent experiments is shown. **g** Hap1-WT and Hap1-SPRING[KO] cells were transduced with a FLAG-tagged constitutively active SREBP2 N-terminal construct. Total cell lysates were immunoblotted as indicated ($N = 2$). All bars and errors represent mean ± SEM; *$p < 0.05$, **$p < 0.01$, ***$p < 0.001$.

and mature (i.e., processed) protein level of SREBP2, and consequentially the level of its target genes in the absence of a change in the level or localization of Site 1 Protease (Supplementary Fig. 2B). We also considered the possibility that proteolytic cleavage by S1P or S2P could be affected by loss of *SPRING*. Similar to SREBPs, the ER-stress-related factor ATF6 undergoes proteolytic processing by these two proteases[30]. We, therefore, tested whether basal- and Tunicamycin-induced ER stress signaling is altered in the absence of SPRING (Supplementary Fig. 2C). Basal ER-stress, as evaluated by expression of ATF6-driven genes and other ER stress-related genes, was not changed in cells lacking SPRING. Importantly, induction of ER stress by Tunicamycin, which promotes ATF6 translocation to the Golgi and subsequent proteolytic processing by S1P/S2P proteases was intact, albeit we did observe a small yet significant reduction. This suggests that the S1P/S2P axis is not globally abrogated in cells lacking SPRING and can still respond to physiological cues.

We also considered the possibility that loss of *SPRING* may result in an intrinsic lesion in SREBP activity (e.g., if it is directly required for transcriptional activation of SREBP). Yet this does not seem to be the case, as when we introduce an N-terminal constitutively active SREBP2 transcriptional domain into Hap1-SPRING[KO] cells we were able to restore the basal and sterol-depletion-induced levels of SQLE (Fig. 2g). To evaluate the consequence of absence of SPRING we compared the transcriptional profile of Hap1-WT and Hap1-SPRING[KO] cells (Fig. 3a and Supplementary Fig. 3B). This comparison confirmed the aberrant activation of the SREBP pathway in Hap1-SPRING[KO] cells. We, therefore, proceeded to compare the global transcriptional response of these cells to sterol-depletion (Fig. 3b, c). While Hap1-WT increased expression of a panel of SREBP-regulated target genes in response to sterol depletion, in the absence of SPRING the expression of these genes was refractory to this treatment. Importantly, the SREBP program could be restored by reintroducing *SPRING* in sterol-depleted Hap1-SPRING[KO] cells (Fig. 3d). Collectively, this set of experiments establishes SPRING as a determinant of SREBP activation in mammalian cells.

**Spring attenuates hepatic SREBP signaling in vivo.** *Spring* is ubiquitously expressed in mouse tissues with slightly higher expression in the liver and kidney (Supplementary Fig. 5A). To study whether our observation in cells extends to the in vivo setting and to investigate the physiological role of Spring, we generated *Spring*$^{(-/-)}$ mice using CRISPR/Cas9 technology. Guide RNAs were designed to target the 5′ region of coding exon 2 and the 3′ region of coding exon 5. We obtained heterozygous mice with a deletion spanning exons 2 until 5 (Fig. 4a). However, heterozygous crosses failed to produce viable homozygous knockout offspring (Fig. 4b). The genetic distribution of offspring was, in fact, consistent with loss of *Spring* being embryonic lethal and suggesting a critical role for *Spring* in mouse development. We confirmed this notion by analyzing embryos obtained in

heterozygous crosses. We did observe abnormally developed embryos at day 7.5 dpc. These embryos were smaller and showed a poorly developed amniotic cavity and allantois in comparison with the adjacent to normally developing embryos in the mouse uterine horn. The abnormal embryos were confirmed to be homozygous null mutants by PCR genotyping (Supplementary Fig. 5B, C).

As we were unable to study constitutive *Spring*$^{(-/-)}$ mice due to early embryonic lethality, we opted to study the consequence of adenoviral-mediated temporal silencing of hepatic *Spring* expression. Following transduction, mice were fasted overnight and subsequently refed for 6 h to maximally induce SREBP signaling. Under this setting, effective *Spring* silencing resulted in significant reduction of *Srebf2* and a panel of its downstream transcriptional targets *Hmgcs*, *Hmgcr*, *Sqs*, *Dhcr24*, *Pcsk9*, and *Ldlr* (Fig. 4c). Notably, *Srebf1* and its target genes *Fasn*, *Scd1*, and *Acc* were unchanged. In this experimental setting no differences were observed in hepatic and plasma lipids. These results provide an indication that *Spring* regulates SREBP signaling in vivo and has a crucial physiological role during embryogenesis.

**SPRING is a glycosylated Golgi-resident membrane protein.** *SPRING* encodes for a 205 amino acid protein with no apparent homology to other proteins and is predicted to have a possible signal peptide, a single transmembrane-spanning domain, and a cysteine-rich motif (Fig. 5a). To gain insight into how SPRING influences SREBP activity we determined its cellular localization. We first considered the possibility that SPRING may be a secreted protein. However, our analysis indicates that this is not the case, but that SPRING is associated with cellular membranes (Supplementary Fig. 6A, B). In transfected HeLa cells SPRING was predominantly co-localized with the Golgi marker GM130 (Fig. 5b). Consistent with glycosylation of SPRING, N-glycosidase (PNGase-F) removed glycans from SPRING protein (Supplementary Fig. 6C). Further, mutation of the single predicated glycosylation site (Asn-67) abolished SPRING glycosylation (Fig. 5c). To interrogate the nature of the glycan chain present on SPRING we used Endoglycosidase-H (Endo-H), which removes glycans from ER- and *cis/medial* Golgi-resident proteins, before they acquire complex modifications[31]. We observed that unlike LDLR, SPRING glycosylation remained Endo-H-sensitive (Supplementary Fig. 6D). Together with the observed localization of SPRING in the Golgi (Fig. 5b), sensitivity to Endo-H suggests that SPRING is present in the *cis/medial* Golgi, which is where the SCAP-SREBP-S1P machinery is located[32,33]. To determine the topology of SPRING in the Golgi membrane we performed protease protection assays. Unlike LDLR-HA, which has its C-terminus exposed to the cytoplasm and is hence sensitive to tryptic digestion, SPRING-V5 was refractory (Fig. 5d). Permeabilization of cellular membranes with Triton X-100 rendered SPRING-V5 sensitive to tryptic digestion. In aggregate with our observation on SPRING glycosylation, these results indicate that the C-terminus of SPRING faces the Golgi lumen.

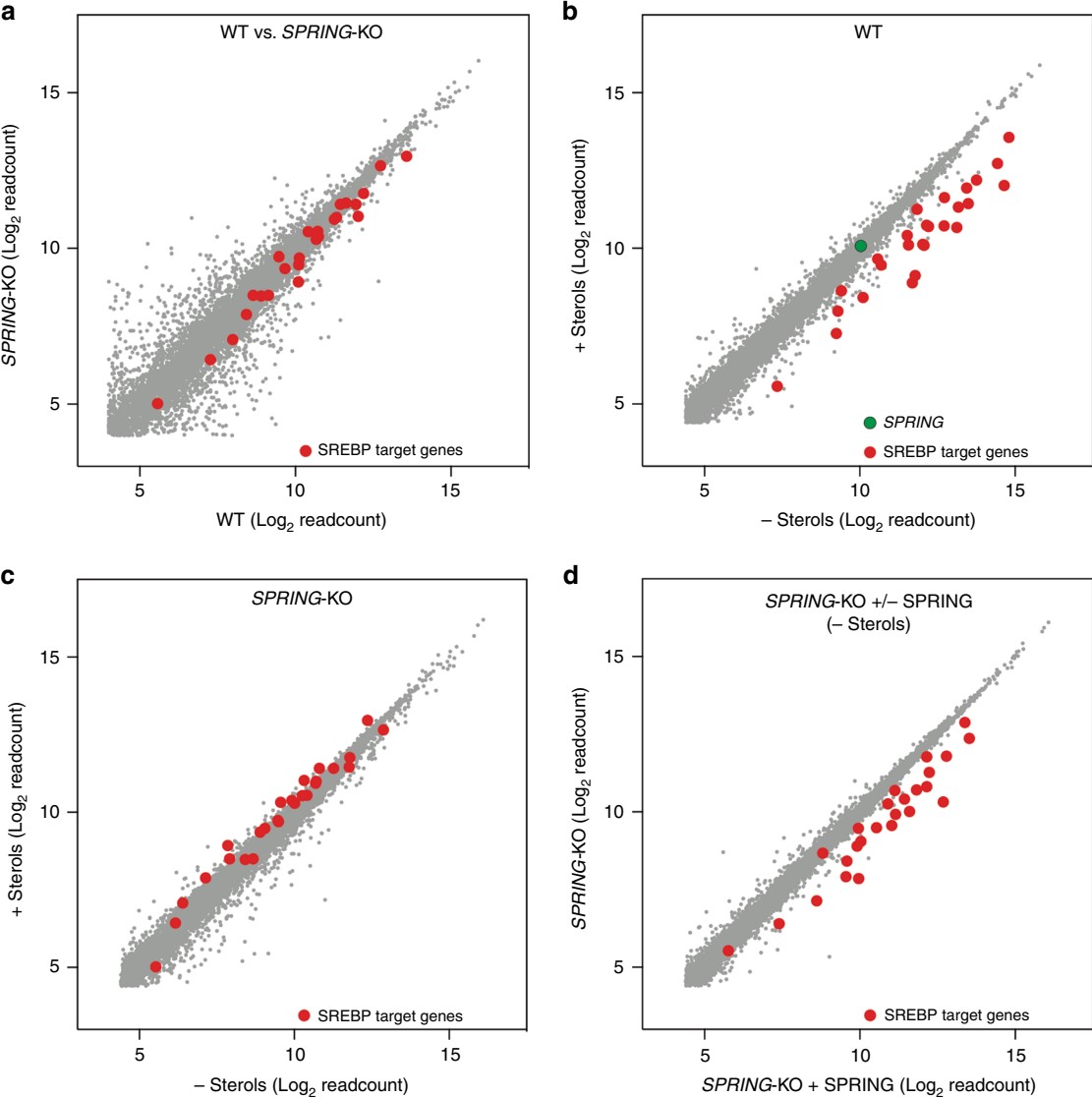

**Fig. 3 Global SREBP-signaling requires SPRING. a–d** Comparison of transcriptional profiles obtained by RNAseq of **a** Hap1-WT and Hap-SPRING[KO] cells, and **b–d** the transcriptional response of the indicated cells to sterol depletion. SREBP target genes (*HMGCR, HMGCS1, SQLE, ACACA, ACAT2, ACSS2, CYP51A1, DHCR24, DHCR7, EBP, FASN, FDFT1, FDPS, GGPS1, GGPS1, HSD17B7, IDI1, INSIG1, LDLR, LSS, MVD, MVK, NSDHL, PCSK9, SC5D, SCD1, SREBF2, TM7SF2*)[4,18] are shown in red and *SPRING* in green.

Intrigued by the fact that cholesterol-sensing by SCAP occurs initially in the ER, we asked whether delivery of SPRING to the Golgi compartment was an absolute requirement for its ability to regulate SREBP activation. To address this, we generated a SPRING expression construct with a C-terminal (i.e., lumen-facing) ER-retention KDEL signal[34] (SPRING[KDEL]). SPRING[KDEL] was largely retained in the ER, as evident by co-localization with the ER marker protein VAMP-associated protein A (VAPA, Fig. 5e). In contrast to WT SPRING, which could rescue sterol-dependent regulation of SQLE when introduced into Hap1-SPRING[KO] cells, SPRING[KDEL] failed to do so (Fig. 5f). Taken together, these results demonstrate that regulation of SREBP activity by SPRING requires its Golgi localization.

**SPRING governs SCAP localization in cells**. Trafficking to and subsequent proteolytic processing of SREBPs in the Golgi is critically dependent on a stoichiometric interaction with SCAP[7,8]. Although the levels of SREBP are drastically decreased, SREBP processing can still occur in SPRING[KO] cells (*c.f.* Fig. 2a, b, f). We, therefore, considered the possibility that SCAP function may be affected by SPRING. Regulation of SCAP retention in the ER requires interaction with INSIGs. This prompted us to investigate whether akin to INSIG, SPRING also interacts with SCAP. We used a co-immunoprecipitation approach and found that SCAP can interact with SPRING when the two are over-expressed in a model system (Fig. 6a). With this assay, we are unable to formally establish the cellular localization of this interaction. However, we point out that using the same approach we were unable to detect an interaction between SPRING and INSIG. The potential functional significance of the SCAP-SPRING interaction was then evaluated in CHO cells that stably produce SCAP-eGFP (CHO-SCAP-eGFP)[35] in which *Spring* was ablated by CRISPR/Cas9-mediated genome editing. In control cells grown in sterol-containing media SCAP-eGFP was predominantly located in the ER and localization markedly shifted towards the Golgi upon sterol-depletion (Fig. 6b and Supplementary Fig. 7A). However, in CHO cells devoid of *Spring*, SCAP-eGFP appeared trapped in

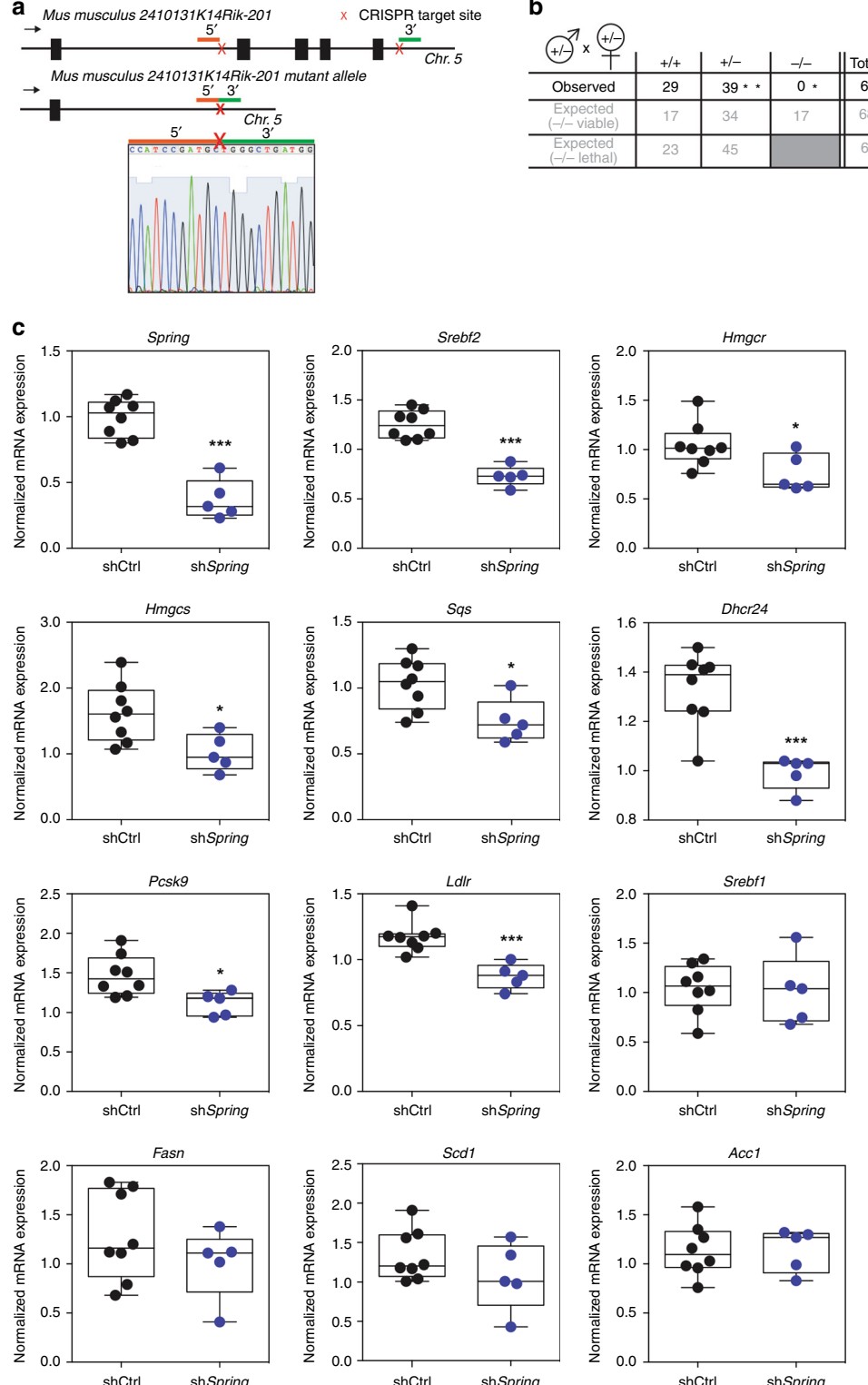

**Fig. 4 SPRING is essential for mouse embryogenesis and for hepatic SREBP signaling. a** Illustration of the genomic organization of murine SPRING (2410131K14Rik-201) and the allele obtained after CRISPR/Cas9 editing. The 5′ and 3′ gRNAs are indicated in red and green, respectively. Sanger sequencing of amplified genomic DNA was performed to confirm the deletion of exons 2–5. **b** Table showing an overview of the obtained genotypes from various crosses of heterozygous mice. *$p < 0.005$, lower than expected by Pearson's Chi-square test with 2 df; **equal to expected when assuming embryonic lethality of homozygous null mice, Chi-square = 2.37, $0.9 > p > 0.1$ with 1 df. **c** WT C57BL/6J mice were administered Ad-shCtrl or Ad-sh*Spring* ($N = 8/5$ animals per group, respectively) via tail-vein injection. After 7 days, mice were fasted overnight and subsequently refed for 6 h. Total liver RNA was isolated and expression of the indicated genes was determined by qPCR. Each individual mouse is plotted within the box and whiskers plot that depicts the median line, the 25th and 75th percentile, and the min-max values. *$p < 0.05$, ***$p < 0.001$.

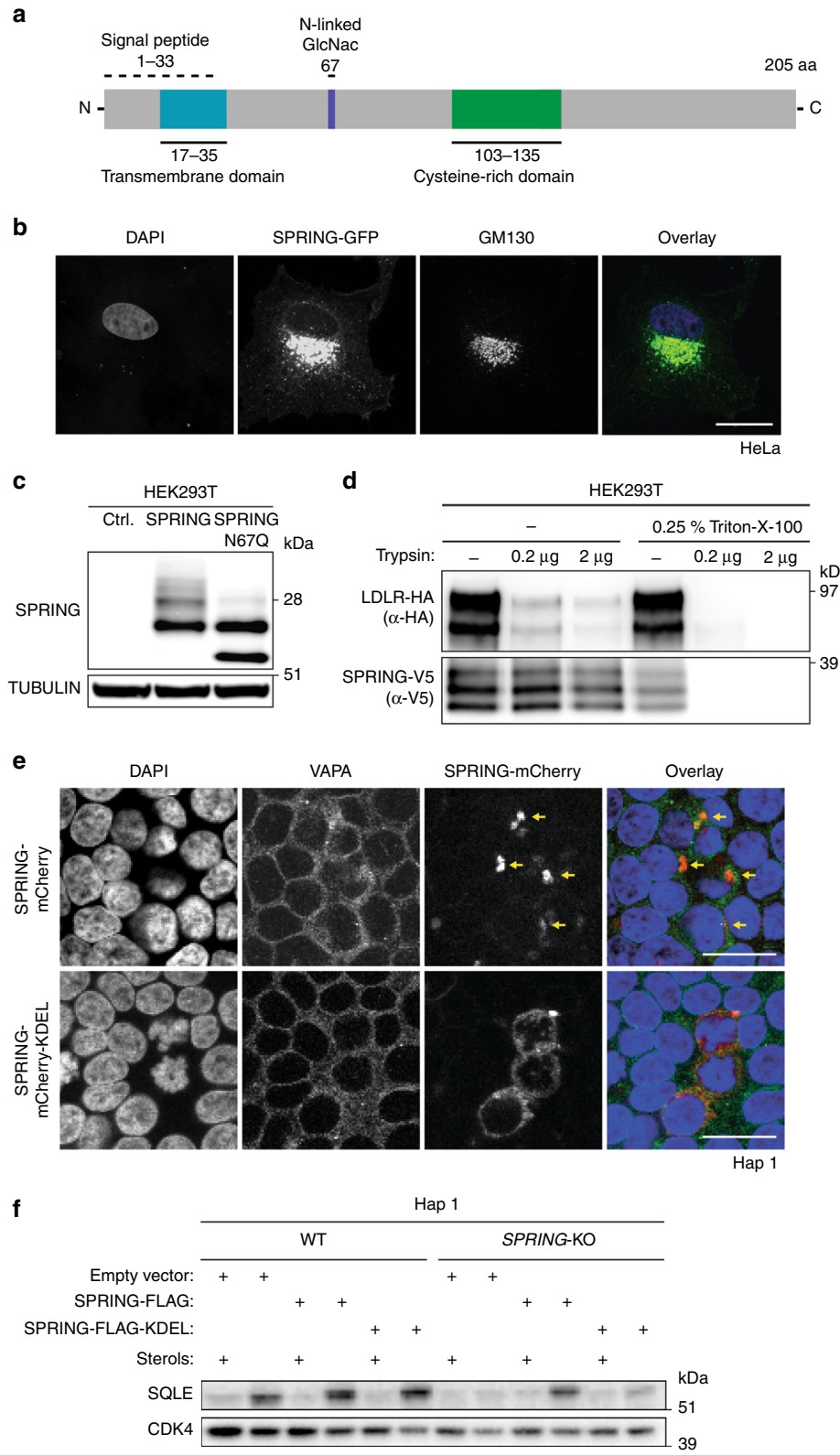

the Golgi irrespective of the cellular sterol status. Importantly, ER-localization of SCAP-eGFP could be restored in the SPRING[KO] cells grown in the presence of sterols by introducing back SPRING expression. Mislocalization of SCAP to the Golgi in SPRING[KO] cells likely results in depletion of functional SCAP in the ER, which is required at stoichiometric levels to support exit

of SREBP towards the Golgi. However, these experiments were conducted with cells that stably over-produce SCAP-eGFP, and are thus highly suitable to track SCAP localization, but may mask effects on SCAP protein level. We. therefore. evaluated endogenous SCAP protein in Hap1 cells (Fig. 6c and Supplementary Fig. 2D). Remarkably, SCAP protein was reduced in Hap1

**Fig. 5 Localization of SPRING to the Golgi is required for regulation of SREBP. a** Schematic depiction of SPRING's predicted domains and post-translational modifications (www.uniprot.org). **b** HeLa cells were transfected with a SPRING-GFP expression construct. Subsequently, cells were counterstained against the Golgi-resident protein GM130. DAPI was used to stain the nuclei; scale bar, 10 μm. **c** HEK293T cells were transfected with expression constructs encoding SPRING WT or a N67Q mutant. Total cell lysates were immunoblotted as indicated ($N = 3$). **d** HEK293T cells were co-transfected with expression constructs encoding LDLR-HA and SPRING-V5. Cellular membranes were isolated and subjected to tryptic digestion in the presence or absence of the permeabilizing detergent Triton-X-100. An equal amount of protein was immunoblotted as indicated. **e** Immunofluorescence staining of Hap1 cells transfected with expression constructs encoding SPRING-mCherry or SPRING-mCherry-KDEL and nuclei were counterstained with DAPI; scale bar, 25 μm. **f** Hap1-WT and Hap-SPRING[KO] cells were transfected as indicated and total cell lysates immunoblotted as shown ($N = 2$).
**b**, **d**, **e** Representative images of three independent experiments are shown.

SPRING[KO] cells. This is consistent with functional SCAP deficiency in these cells, and implies that increasing the level of SCAP should overcome the SREBP-signaling defect in SPRING[KO] cells. In line with this idea we found that introducing *SCAP* into Hap1 SPRING[KO] cells, similar to introducing back *SPRING*, fully restored the SREBP2-mediated sterol-dependent response (Fig. 6d). Functionally, introduction of SCAP was sufficient to restore LDL uptake and de novo cholesterol biosynthesis (Fig. 6e, f and Supplementary Fig. 7B).

Finally, it is well recognized that intact SREBP signaling is required for cell growth and proliferation. Moreover, there is increasing evidence that cancer cells activate SREBP signaling as a means to produce lipids to support their rapid growth[3,36]. We, therefore, interrogated the Dependency Map (DepMap, www.depmap.org) repository which aims to identify genetic vulnerabilities in human cancer[37,38]. Within the database of 342 cancer cell lines (27 distinct lineages) that were subjected to genome-wide CRISPR/Cas9 lethality, *SPRING* expression emerged as a selective gene in 337/342 of the evaluated cell lines (Supplementary Fig. 8A). This suggests that a wide-variety of tumor cells is dependent on *SPRING* expression for growth. Remarkably, in this panel of cell lines the top co-dependent genes with *SPRING* were those associated with the core SREBP machinery (Supplementary Fig. 8B), mirroring our haploid genetic screen results. This observation lends further support for a central role for SPRING in regulating the SREBP pathway and its potential role in proliferative diseases. Collectively, our in vitro and in vivo findings support the idea that SPRING is a Golgi-resident factor required for maintaining SCAP function, and that loss of *SPRING* results in functional depletion of SCAP in the ER and attenuation of SREBP signaling (Supplementary Fig. 9).

## Discussion

The SREBP transcriptional network is a central determinant of homeostatic lipid metabolism. Dysregulation of this pathway underlies development of human conditions, exemplified by development of hypercholesterolemia and ensuing coronary artery disease due to mutations in the SREBP-regulated gene *LDLR*[25]. Therefore, elucidating the mechanisms that govern the SREBP pathway is of outmost importance. Genetic approaches have been paramount in clarifying the molecular components that control cholesterol and fatty acid metabolism that are regulated by SREBPs[26]. Mammalian haploid genetic screens have been applied to interrogate a variety of cellular processes and phenotypes, amongst others, pathogen entry[27,39–42], signal transduction[24], modes of toxin action[43], and gene essentiality[28]. Yet this methodology is also well suitable to address lipid-associated questions, and accordingly we recently applied this approach to investigate the control of sterol-stimulated degradation of HMGCR[44]. In this study, by combining three independent SREBP-related screens we identify *SPRING* (*C12ORF49*) as a previously uncharacterized regulator of the SREBP pathway.

SPRING is a Golgi-resident glycosylated membrane protein and together with its orthologs forms an uncharacterized protein family (Pfam UPF0454), which bears no substantial homology with other human proteins. We found that the primary phenotype of cells lacking *SPRING* is decreased basal levels of precursor and mature SREBPs and SREBP signaling, and an inability to enhance SREBP signaling so as to mount a homeostatic response to sterol-depletion. This is highly reminiscent of what is observed in cells lacking SCAP[45] and in liver-specific *Scap* knockout mice[46]. Accordingly, we have narrowed the primary lesion in SREBP signaling in SPRING[KO] cells to SCAP functionality. Namely, in SPRING[KO] cells SCAP levels are reduced and the protein is trapped in the Golgi irrespective of the cellular sterol status. Consequentially, ectopic over-expression of SCAP rescues SREBP signaling in SPRING[KO] cells, in line with functional depletion of SCAP in the ER. The COPII-mediated anterograde transport of SCAP from the ER to the Golgi is well-studied[2,32,47]. However, despite being an essential part of SCAP's life cycle, the molecular determinants and events that govern its Golgi activity and eventual retrograde COPI-mediated trafficking back to the ER have received only limited attention[19,48].

Upon delivery to the Golgi, the SCAP-SREBP complex is tethered to PAQR3[49]. The interaction between SCAP and PAQR3 anchors the complex to the Golgi and is necessary to support proteolytic activation of SREBP and hepatic lipid synthesis. Cleavage by S1P is also a perquisite to release SCAP for COPI-mediated retrograde transport to the ER, as pharmacological or genetic inhibition of this process prevents SCAP retrograde trafficking, and instead directs SCAP towards lysosomal degradation[48]. The phenotype of SPRING[KO] cells is consistent with the potential involvement of SPRING in the process of retrograde transport of SCAP. Retrieval of proteins to the ER is classically dependent on the presence of a C-terminus -KKXX or -KDEL sequence[34], both of which are absent in SCAP. Yet as these are also absent in SPRING we find it unlikely that SPRING directly acts as a retrieval adaptor protein. Alternatively, it is possible that SPRING is a licensing factor that is required for directing SCAP towards retrograde transport, possibly by releasing it from PAQR3 or other retention signals. However, PAQR3 was not identified in our screens, possibly reflecting functional redundancy between the 11 PAQR family members[50], or cell-type specific differences. It is also formally possible that SPRING governs SCAP by controlling the fraction that is directed towards the degradative pathway. Absence of SPRING, akin to preventing S1P-dependent cleavage of SREBP[48], could result in functional depletion of SCAP. Our observation that the ATF6-mediated stress response - which like SREBP activation requires the sequential proteolytic processing by S1P and S2P yet does not require SCAP - is also reduced in the absence of *SPRING* provides support for this scenario and may explain reduced SCAP protein level in SPRING[KO] cells. As such, SPRING could influence additional processes beyond modulating SCAP and the SREBP pathway. Finally, while speculative, it is intriguing to consider a potential role for SPRING's cysteine-rich motif in its function. Notably, the cysteine-rich domain present in the ectodomain of the Hedgehog receptor Smoothened was recently reported to

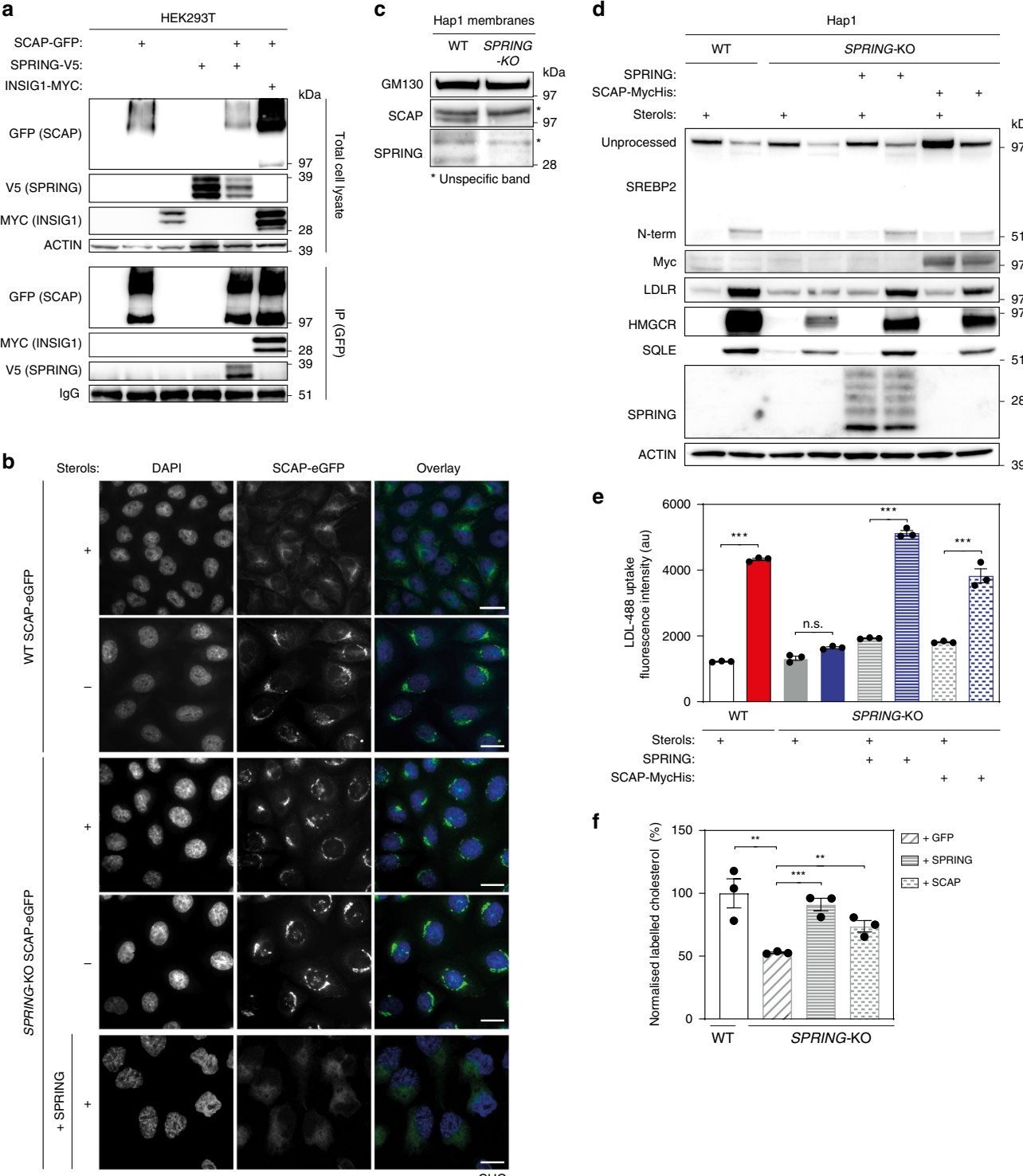

**Fig. 6 SPRING modulates SCAP function. a** HEK293T cells were transfected with the indicated expression constructs. Total cell lysates and immunoprecipitated fractions were analyzed by immunoblotting as indicated ($N = 3$). While we determined that SPRING was detected in the co-IP fraction with SCAP, its level was not enriched in this fraction relative to the level in total cells. **b** Representative fluorescence images of CHO-SCAP-eGFP-WT and CHO-SCAP-eGFP-SPRING[KO] cells cultured in the presence or absence of sterols; scale bar, 10 μm. See also Supplementary Fig. 7A. Representative images of three independent experiments are shown. **c** An equal amount of crude membrane fractions from Hap1-WT and Hap1-SPRING[KO] cells were immunoblotted as indicated ($N = 4$), with GM130 serving as loading control. **d, e** Hap1-WT and Hap-SPRING[KO] cells that stably express *SPRING* or *SCAP* were treated as indicated and **d** total cell lysates were immunoblotted as indicated ($N = 3$), and **e** fluorescent LDL uptake was measured by FACS ($N = 3$ biologically independent samples). Note that the first 4 bars are from Fig. 2d and are shown for comparison. **f** Hap1-WT and Hap-SPRING[KO] cells that stably express *SPRING, SCAP*, or *GFP* were cultured for 24 h in medium containing 10% LPDS, 3 mM β-methyl-cyclodextrin, and ${}^{13}C_2$-Sodium acetate ($N = 3$ biologically independent samples/group). Following lysis incorporation of ${}^{13}C_2$-acetate into cholesterol was determined by mass-spectrometry as described in the Methods section. All bars and errors represent mean ± SEM; *$p < 0.05$, ***$p < 0.001$.

directly bind cholesterol, and this was sufficient to induce receptor activation[51,52]. While the Smoothened and SPRING cysteine-rich domains differ in the number and organization of cysteines, it is possible that binding of cholesterol, or a related sterol, to this motif is required for regulation of SPRING function. Possibly, binding of a sterol to this domain may result in a regulatory conformational change akin to that occurring in SCAP[11]. Evidently, addressing these possibilities and the detailed mechanism by which SPRING regulates the SREBP pathway will require the development of novel sensitive and quantitative assays to monitor, amongst others, SCAP trafficking.

To demonstrate that our findings in cells extend to a physiological in vivo setting we developed *Spring* knockout mice. Genetic ablation of *Spring* resulted in early embryonic lethality demonstrating that Spring is required for embryonic development in mice. The embryonic lethality associated with *Spring* ablation is similar to that observed in mouse models of *Srebf1* and *Srebf2* deletion[53,54], albeit in these models embryonic lethality is not absolute. Whether the lethality observed in $Spring^{(-/-)}$ embryos is related to its role in lipid metabolism remains to be investigated. Using adenoviral-mediated silencing of hepatic *Spring* expression we found marked attenuation of SREBP2 signaling in the liver, with limited effect on the SREBP1 pathway. The effect on SREBP2 is similar to the phenotype observed in liver- or intestine-specific *Scap* knockout mice[46,55], but the latter was somewhat unexpected given our results in primary hepatocytes and cell lines, and the functional requirement of an intact SREBP2 pathway to drive the SREBP1-controlled genetic program[53,56]. This could possibly be due to the short-term duration of the experiment, to the presence of residual *Spring* mRNA expression, or the lack of a dietary challenge. Nevertheless, these results demonstrate that *Spring* is a regulator of the SREBP program in vivo in mice.

Finally, genetic variation in components of the SREBP genetic program, such as in the *LDLR*, *PCSK9*, and *HMGCR*, is associated with lipid traits in humans[57]. It remains to be seen whether genetic variation in *SPRING* contributes to lipid traits and associated diseases in humans. Our bioinformatic analysis suggests that a wide-variety of human tumor cells display dependency on expression of a core set of SREBP-centered genes, including *SPRING*, for their growth thereby expanding the potential involvement of SPRING to other lipid-associated conditions. In conclusion, we report here the identification of SPRING as a previously unrecognized regulator of the SREBP program. Our in vitro and in vivo findings warrant further studies to evaluate the contribution of SPRING to lipid metabolism.

## Methods

**Chemicals**. Simvastatin sodium salt was purchased from Calbiochem. All other reagents (including $^{13}C_2$ sodium acetate and methyl-β-cyclodextrin) were purchased from Sigma.

**Cell lines and cell culture**. HeLa, Hepa1-6, and HEK293T cells were obtained from the ATCC and cultured in Dulbecco's modified Eagle's medium (DMEM) supplemented with 10% fetal bovine serum (FBS) and 10,000 U/mL penicillin–streptomycin (Gibco) in a humidified atmosphere at 37 °C and 5% $CO_2$. Hap1 cells were cultured similarly but in Iscove's modified Eagle's medium (IMDM). CHO-SCAP-eGFP cells (a kind gift from Peter Espenshade, John Hopkins, USA) were cultured in a 1:1 mixture of Ham's F-12 medium and DMEM supplemented with 5% FBS and 10,000 U/mL penicillin–streptomycin in a humidified atmosphere at 37 °C and 8% $CO_2$, as previously described[35,58]. For sterol-depletion cells were cultured in sterol-depletion medium (DMEM, IMDM, or DMEM/F12) supplemented with 10% lipoprotein-deficient serum (LPDS), 2.5 μg/mL simvastatin, and 100 μM mevalonate as indicated and previously described[59]. For specifically evaluating cholesterol synthesis, cells were cultured in medium supplemented with 10% LPDS and 3 mM β-methyl-cyclodextrin and cell viability was monitored using the MTT assay. To generate Hepa1-6 murine hepatocytes that stably produce Cas9 (Hepa1-6-Cas9), cells were transduced with a lentiviral construct encoding Cas9 and subsequently individual clones were selected and expression of Cas9 verified. Mouse hepatocytes expressing Cas9 were isolated

from Cas9 knock-in mice[60,61] (#02857, The Jackson Laboratory) and cultured as described previously[61]. Hepatocytes from wildtype (WT) littermates were used as control WT cells.

**Generation of FASN^KO and SPRING^KO cells**. To ablate expression of *FASN* and *SPRING* in human cell lines we used CRISPR/Cas9-mediated genome editing as previously described[62]. Briefly, guide RNAs (sgRNAs) targeting an exon-coding region of the respective each gene were designed and cloned into px330 (Addgene #42230). The sequences of the sgRNAs are shown in Supplementary Data 1. Subsequently, cells were transfected and independent clones obtained after selection. To ablate *Spring* in CHO-SCAP-eGFP cells sgRNAs targeting hamster *Spring* or the safe-harbor locus *Ppp1R12c* were cloned into lentiCRISPRv2 (Addgene #52961). Lentiviral particles were produced in HEK293T cells and used to transduce and select individual clones CHO-SCAP-eGFP clones. Proper genome editing in all individual clones was confirmed by sequencing. To target *Spring* in mouse primary WT or Cas9 expressing hepatocytes, cells were isolated and cultured as described above. Cells were infected 4 h post isolation with Ad-3xsgRNA-*Spring* at an MOI of 20 for 24 h. Subsequently, cells were washed and sterol-depleted for 16 h after which cells were harvested for immunoblotting and gene expression analysis. To ablate *Spring* in Hepa1-6-Cas9 cells were infected similarly at an MOI of 50 for 96 h.

**Generation of Hap1 SQLE-mNeon cells**. To insert the mNeon-2A-PURO reporter cassette into the endogenous *SQLE* locus we used microhomology-based CRISPR/Cas9-CRIS-PITCh methodology[63]. Briefly, this technique allows integration of a mNeon-2A-PURO cassette into the ultimate coding exon of *SQLE* and subsequent selection of puromycin resistant clones, as we have recently reported for *HMGCR*[44]. The donor fragment containing the microhomology and mNeon-2A-PURO sequences, as well as the sgRNA sequences are shown in Supplementary Data 1 and 2. Independent clones were expanded and genome editing was confirmed by sequencing, immunoblotting, and immunofluorescence.

**Human haploid genetic screens**. *FASN fitness screen.* Hap1 FASN^KO cells were mutagenized as previously described[28]. In brief, retroviral gene-trap virus was produced in HEK293T cells. 40 million Hap1 FASN^KO cells were transduced using virus combined from multiple harvests on 3 consecutive days. The obtained mutagenized library was cultured for 12–14 days while maintaining at least 4-fold library complexity. Afterward, cells were fixed with BD Fix buffer I (BD Biosciences) and stained for DNA content with 5 μg/mL propidium iodide (PI). To ensure that only gene-trap insertions in haploid cells are analyzed, cells were sorted by FACS for G1 phase. Genomic DNA was isolated using the Qiagen DNA isolation kit. Insertions were mapped according to the protocol described in Blomen et al.[28].

*SQLE-mNeon screen.* In order to identify regulators of SQLE we prepared a library of mutagenized Hap1-SQLE-mNeon cells using a gene-trap retrovirus expressing blue fluorescent protein (BFP), as described previously[28,64]. Briefly, 5 × $10^8$ Hap1-SQLE-mNeon cells were seeded and transduced with virus from two combined harvests on three consecutive days in the presence of 8 μg/mL protamine sulfate (Sigma). Mutagenized cells were expanded to thirty T175 flasks at a confluence of ~80%. Subsequently, cells were cultured in sterol-depletion medium for a total of 24 h and with 50 μg/ml β-methyl-cyclodextrin-cholesterol (Sigma) during the last 6 h to stimulate cholesterol-dependent degradation of SQLE-mNeon. At the end of this treatment. the cells were washed twice with PBS, dissociated with TrypLE (Thermo Fisher), pelleted, and fixed with BD Fix Buffer I (BD biosciences) for 10 min at 37 °C. After washing twice with PBS containing 10% FCS, the cells were filtered through a 40 μm strainer (BD FalconTM) before sorting two populations of cells (i.e. SQLE-mNeon^LOW and SQLE-mNeon^HIGH) that represent ~5% of the lowest and highest SQLE-mNeon expressing cells from the total cell population, respectively. In addition, in order to reduce potential confounding effects of diploid cells, which are heterozygous for alleles carrying gene-trap integrations, the cells were sorted in parallel for haploid DNA content (G1 phase) by staining with propidium iodide. Cell sorting was carried out on a Biorad S3 Cell sorter until ~10 million cells of each population were collected. Sorted cells were pelleted and genomic DNA was isolated using a DNA mini kit (Qiagen). To assist de-crosslinking of genomic DNA the cell pellets were resuspended in PBS supplemented with Proteinase K (Qiagen) followed by overnight incubation at 56 °C with lysis buffer AL (Qiagen) with continuous agitation. Gene-trap insertion sites of each sorted cell population were amplified using a Linear Amplification polymerase chain reaction (LAM-PCR) on the total yield of isolated genomic DNA[28]. Samples were subsequently submitted for deep sequencing and insertion sites were mapped and analyzed as previously described[44,64].

**Molecular cloning and generation of adenoviral particles**. The full human *SPRING* open reading frame was amplified from Hap1 and HepG2 cDNA (RefSeq NM_024738) and sub-cloned into pDONR221 (Invitrogen) or pENTR1A-GFP (Invitrogen) to create pDONR221-SPRING and pENTR-SPRING-GFP, respectively. The resulting entry constructs were used to generate pLenti6.3-SPRING and pLenti6.3-SPRING-GFP following LR gateway recombination with pLenti6.3-

DEST (Invitrogen). Site directed mutagenesis was used to introduce an N67Q mutation in pDONR221-SPRING. pDONR221-SPRING and pDONR221-SPRING$_{N67Q}$ were used to generate pDEST47-SPRING and pDEST47-SPRING$_{N67Q}$, respectively. *SPRING* cDNA was also cloned into the retroviral vector pBABE-PURO (Addgene #1764). mCherry (derived from pmCherry-C1 (Clontech)) was cloned at the C-terminus of SPRING (separated by an alanine linker) to create pBABE-SPRING-mCherry. Additionally, a C-terminal FLAG tag was added to pBABE-SPRING (separated by a AAV linker) to create pBABE-SPRING-FLAG. A KDEL retention signal (5'-AAGGACGAGTTG-3') was added to the C-terminus of pBABE-SPRING-mCherry and pBABE-SPRING-FLAG to create pBABE-SPRING-mCherry$^{KDEL}$ and pBABE-SPRING-FLAG$^{KDEL}$, respectively. The pcDNA-SCAP-GFP and pcDNA4-SCAP-MycHis plasmids were a kind gift from Andrew Brown[65] (Sydney University, Australia). SCAP-MycHis was amplified from pcDNA4-SCAP-MycHis, cloned into pENTR1A (Invitrogen), and subsequently recombined into pLenti6.3-DEST (Invitrogen) using Gateway cloning. All plasmids were verified by sequencing and transfected into cells using JetPrime reagent (Polyplus) according to the manufacturer's protocol. To generate Ad-3xsgRNA-Spring adenoviral particles a geneblock containing 3 guide RNAs targeting murine *Spring* under the control of U6 promoters was ordered (Invitrogen, Supplementary Data 3). The geneblock fragment was cloned into pDONR221 (Invitrogen) and recombined into pAD/PL-DEST (Invitrogen) using Gateway recombination. To generate pAD-sh*Spring* oligonucleotides targeting 3 different coding regions of *Spring* were cloned into pTER+/pENTR (Addgene #430-1) that has been modified by addition of a CMV-GFP cassette (Supplementary Data 1). The resulting pTER+/pENTR-GFP-sh*Spring* or pTER+/pENTR-GFP-shScrambled were recombined into pAD-BLOCK-iT (Invitrogen) using Gateway cloning, tested, and the most effective one used. Adenoviral particles were generated and evaluated as previously reported[59], and amplified, purified and tittered (Viraquest).

**Immunofluorescence.** Hap1 and HeLa cells were seeded on poly-L-lysine (Sigma Aldrich) coated cover slips. After indicated treatments, cells were washed three times with ice-cold PBS and fixed using 4% paraformaldehyde (Sigma Aldrich F8775). Cells were permeabilized in 0.1% Triton X-100 (Sigma), blocked with 5% BSA (Sigma) and stained with the following primary antibody for 16 h: VAPA (a kind gift from Sjaak Neefjes, Leiden University Medical Center, the Netherlands, 1:200), and GM130 (Cell Signaling 12480, 1:1000). Secondary antibodies conjugated with Alexa-Fluor-568 or Alexa-Fluor-488 were used (Thermo Fisher, 1:400). CHO-SCAP-eGFP cells were plated as above and transfected with Calnexin-mCherry and Giantin-mScarlet expression plasmids to label the ER and Golgi compartments, respectively (plasmids were a kind gift from Eric Reits and Joachim Goedhart, University of Amsterdam). Cells were subsequently treated as indicated. DAPI was added during incubation with the secondary antibodies in 5% BSA/PBS for 1 hr at room temperature. Cover slips where washed three times for 20 minutes and mounted on standard glass slides. Imaging was performed on a Leica SP5 confocal microscope using LCS software.

**Immunoblotting, deglycosylation assay, and co-IP.** Total cell lysates were prepared in RIPA buffer (150 mM NaCl, 1% NP-40, 0.1% sodium deoxycholate, 0.1% SDS, 100 mM Tris-HCl, pH 7.4) supplemented with protease inhibitors (Roche). Lysates were cleared by centrifugation at 4 °C for 10 min at 10,000 × g. Subsequently, cleared lysates were separated on NuPAGE Novex 4–12% Bis-Tris gels (Invitrogen) and transferred to nitrocellulose membranes. Membranes were probed with primary antibodies, which are listed in Supplementary Data 4. Secondary horseradish peroxidase-conjugated antibodies (Invitrogen) were used and visualized with chemiluminescence on a LAS4000 (GE Healthcare). In some experiments proteins were fractionated before immunoblotting as previously reported[66]. For evaluating N-glycosylation, dithiothreitol (DTT) was added to total cell lysates to a final concentration of 50 mM. Subsequently, lysates were heated to 95 °C for 5 min and after cooling recombinant Peptide-N-Glycosidase F (PNGase-F) (Promega) was added at a concentration of 1 unit/μL. The mixture was incubated for 3 h at 37 °C before analysis by immunoblotting. Similar de-glycosylation assays using Endoglycosidase H (EndoH) (NEB) were done according to the manufacturer's protocol. For co-immunoprecipitation (Co-IP) experiments, total cell lysates were prepared in NP-40 buffer (150 mM NaCl, 5 mM EDTA, 1% NP-40, 50 mM Tris-HCl, pH 7.4) supplemented with protease inhibitors (Roche) and incubated overnight at 4 °C with the indicated antibodies and protein A/G-magnetic Dynabeads (Thermo Fisher) according to the manufacturer's instructions. Beads were collected using a DynaMag magnet (Thermo Fisher) and washed three times with lysis buffer. Bound proteins were eluted and analyzed by immunoblotting. As loading controls Actin (1:3000), GM130 (1:1000) or Cdk4 (1:1000) were used and all shown immunoblots are representative of at least 3 independent experiments unless otherwise indicated.

**Protease protection assay.** To determine SPRING topology, an equal amount of isolated cellular membranes was treated with trypsin as described previously[67]. Briefly, membranes were incubated with the indicated amounts of trypsin in the presence or absence of Triton X-100 for 30 min at 30 °C. Reactions were stopped by the addition of loading buffer and heat inactivation at 95 °C for 10 min. Subsequently, samples were analyzed by immunoblotting as indicated above.

**RNA isolation and qPCR.** Total RNA was isolated from cells using a Direct-zol RNA MiniPrep kit (Zymo Research). One microgram of total RNA was reverse transcribed using a cDNA synthesis kit (Biotool). SensiFAST SYBR (Bioline) was used for real-time quantitative qPCR (qPCR) performed on a LightCycler 480 II system (Roche). Gene expression levels were normalized to the expression level of 36B4. Primer sequences are listed in Supplementary Data 5.

**RNAseq analysis.** Two biological replicates of mutant cell lines were analyzed using standard RNAseq methodology. Briefly, cells were cultured in complete medium or sterol-depleted for 16 h, and then scraped in TriZol. A sequencing library was prepared according to standard Illumina RNA-seq protocols. Libraries were multiplexed, clustered and sequenced on an Illumina HiSeq 2500. To visualize data, reads were normalized, averaged and displayed using PRISM software.

**Mouse experiments.** C57BL/6J mice (Charles River) and Rosa26-LSL-Cas9 knock-in mice (#02857, The Jackson Laboratory) were fed a standard chow diet and housed in a temperature-controlled room under a 12-hour light-dark cycle under pathogen-free conditions. For adenoviral infections, age-matched (8–10 weeks old) male mice were injected with $3 \times 10^9$ PFU by tail-vein injection. 8 days later, mice were fasted (overnight) for 16 h and then refed a standard chow diet for an additional 6 h. At the time of sacrifice, liver tissue was collected and immediately frozen in liquid nitrogen and stored at −80°. Liver tissue was processed for isolation of RNA and protein as described above. To assess the tissue distribution of *Spring*, the indicated tissues were collected from 3 individual 12 wk old male C57BL/6J mice. All handling of mice was according to institutional AMC guidelines and regulations and approved by the local ethics committee. The generation of *Spring* knock-out mice was performed under the supervision and with the approval of the animal committee at The Netherlands Cancer Institute and comply with local and international regulations and ethical guidelines. Briefly, Cas9 mRNA and two gRNAs targeting the 5' of exon 2 (5'-TGCCATCCGATGCAAT GCGCA**GG**-3') and the 3' of exon 5' (5'-AGGCAAGTTGGGCGTACTGC**TGG**-3') were injected in zygotes. To identify the edited allele PCR was performed using primers Fw: 5'-CCCAGATTGCCTTCCCACAG-3' and Rv 5'- ATTACGCTGTG ATCCCCACA-3'. For genotyping and phenotyping of single embryos, 7.5 dpc (days postcoitum) embryos were obtained together with the uterine horns. The tissues were fixed in EAF (ethanol/acetic acid/formaldehyde) and embedded in paraffin, from which 2 μm- and 10 μm-thick sections were made for pathologic analysis, and for genomic DNA isolation by laser-guided microdissection for PCR analysis, respectively. All animal procedures (handling, experiments, and generation of *Spring*$^{-/-}$ mice) were approved by the relevant ethics committee.

**LDL uptake assay.** The production of DyLight488-labeled LDL and LDL uptake assays were done as previously described[68]. Briefly, Hap1 cells were plated and washed twice with PBS on the following day. Subsequently, LDL uptake was initiated by incubating cells with 5 μg/ml DyLight488-labeled LDL in IMDM supplemented with 0.5% BSA for 1 h at 37 °C. Cells were then washed twice with PBS supplemented with 0.5% BSA, dissociated, and resuspended in FACS buffer (2 mM EDTA, 0.5% BSA in PBS). Cells were then fixed with 4% paraformaldehyde and cellular LDL uptake was determined by flow cytometry on a CytoFLEX Flow cytometer (Beckman Coulter). Intact cells were gated by standard FSC vs SSC gating

**Surface LDLR assay.** To determine the level of surface LDLR, cells were treated as indicated in the figure legends, dissociated using Accutase (STEMCELL technologies), and washed once with FACS buffer (2 mM EDTA, 0.5% BSA in PBS). Subsequently, cells were stained with an Allophycocyanin (APC)-conjugated mouse anti-human LDLR antibody (R&D; #472413, 10 μl/$1 \times 10^6$ cells) according to the manufacturer's instructions. Subsequently, cells were washed three times with FACS buffer, fixed with 4% paraformaldehyde (PFA) and analyzed on a CytoFLEX Flow cytometer (Beckman Coulter). Intact cells were gated by standard FSC vs SSC gating. Relative surface LDLR levels were calculated from mean values after correction for background signal.

**Cholesterol synthesis assay.** Hap1 control and Hap1-SPRING$^{KO}$ cells were cultured in 6-well plates in IMDM supplemented with 10% LPDS, 3 mM β-methyl-cyclodextrin and 5 mM $^{13}C_2$-Sodium Acetate for 24 h. Cells were then washed twice with ice-cold PBS, once with 0.9% NaCl followed by addition of 1 mL of ice-cold methanol. For the extraction of sterols cells were scraped and transferred to a 2 mL tube, sonicated with a tip sonicator at 8 Watt and 40 Joule and centrifuged at 14,000 × g for 10 minutes at 4 °C. The supernatant was transferred to a new 1.5 mL tube and evaporated under nitrogen. The dried extract was dissolved in 100 μL methanol and analyzed by an ultra-high-pressure liquid chromatography system (Thermo Scientific) with an Acquity UPLC HSS T3, 1.8 μm particle diameter (Waters, Milford Massachusetts, USA) coupled to a Thermo Q Exactive Plus Orbitrap mass spectrometer with an atmospheric-pressure chemical ionization (APCI) source. The column was kept at 30 °C and the flow rate was 0.2 mL/min. The mobile phase was composed of 100% methanol and the gradient was isocratic for a total runtime of 15 min. Data was acquired in full-scan positive ionization mode. Data interpretation was performed using the Xcalibur software (Thermo

Scientific). $^{13}$C enrichment was calculated based on mass distribution isotopomer analysis (MIDA) and all results were corrected for natural $^{13}$C abundance as described[69].

**Statistics**. Statistical analyses were performed with the Prism software package V6. Results were evaluated by two-sided Student's *t*-test when comparing two groups. When 2 or more groups were compared One-way analysis of variance (ANOVA) was used in combination with Tukey's test for multiple comparisons. For knockout mice, the significance of the observed frequencies of genotypes in relation to the expected numbers was analyzed by Pearson's Chi-square test. In the adenoviral experiment, one mouse was excluded as the majority of the measured values were identified as outliers (Grubbs test, α = 0.1). All measurements were taken from distinct samples. SEM is indicated by error bars and *p* values are indicated by asterisks: \**p* < 0.05, \*\**p* < 0.01 and \*\*\**p* < 0.001.

**Reporting summary**. Further information on research design is available in the Nature Research Reporting Summary linked to this article.

## Data availability

Source data for the figures are provided with the paper, and additional data are available from the corresponding author upon reasonable request. RNAseq datasets are available from GEO (GSE143530). The genetic screens are available at https://phenosaurus.nki.nl and the corresponding raw DNA datasets are available from SRA (BioProject ID: 596944). Source image data underlying Figs. 1b, 2b, e, 5c, d, 6a, c, and d and Supplementary Figs. 1B, 2B, D, 4B, C, 6A, B, C and D are provided as a Source Data File. The numerical data underlying Figs. 2a, c, d, e, 4c, 6e, and f and Supplementary Figs. 2C, D, 4A are provided as a Source Data File.

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

## Acknowledgements
We thank members of the Zelcer and Brummelkamp groups, Irith Koster and Ru Koster for their comments, and Niels Kloosterhuis for assistance with isolation of primary hepatocytes. AL is supported by a Dekker grant from the Dutch Heart Foundation (2016T015). JT is supported by an AMC PhD fellowship. TB is supported by the Oncode Institute and the Cancer Genomics Center (GCG.nl), a Vici grant from the Netherlands Organization for Scientific Research (NWO; 016.170.033), a grant from the KWF (NKI-2015–7609), and an Ammodo KNAW Award 2015. NZ is an Established Investigator of the Dutch Heart Foundation (2013T111) and is supported by a research grant from the Ara Parseghian Medical Research Fund, an ERC Consolidator grant (617376) from the European Research Council and by a Vici grant from the Netherlands Organization for Scientific Research (NWO; 016.176.643).

## Author contributions
The study was jointly conceived and designed by T.B. and N.Z. All authors contributed extensively to the work presented in this paper. A.L., M.R., J.N., L.T.J., J.T., S.H., M.v.d.B., and S.S. designed, performed, collected data and analyzed the cell-based experiments. M.R., J.N., and L.T.J. performed and analyzed the RNAseq experiments. M.v.W. measured and analyzed cholesterol synthesis. A.L., L.v.d.H., J.S., I.H., L.K., and R.O., and B.v.d.S. designed, performed, collected data and analyzed the mouse-based experiments. A.L., J.N., T.B., and N.Z. wrote the paper. All authors discussed the results and implications and commented on the paper.

## Competing interests
The authors declare no competing interests.
