## [Peer Review File · Nature Communications]

Reviewers' comments:

Reviewer #1 (Remarks to the Author):

In this study, the authors use a clever screening strategy using three independent approaches to uncover SPRING as a new factor involved in SREBP signaling and cholesterol metabolism. The screen identified known modulators of SREBPs – Scap, Site-1 protease, Site-2 protease – and a new gene, SPRING. Any new modulator of the SREBP pathway would be a major advance, therefore the finding of SPRING could be a big step forward.

Unfortunately, the effects of knockdown or selective deletion of SPRING on SREBP signaling are quite modest (Figure 2D). Moreover, studies to elucidate the mechanism by which SPRING affects SREBP are not convincing. The authors claim that SPRING is a Golgi-resident protein that facilitates the recycling of Scap from the Golgi to the ER so that Scap can carry out another round of SREBP transport. The experiment where ER-localized SPRING-KDEL lowers levels of squalene epoxidase (a proxy for transport of Scap and SREBP) is clever. However, many of the authors findings could be attributed to lower levels of Scap, so their hypothesis is intriguing but not proven. Much more cell biology and biochemistry would need to be done to establish their mechanistic model. Until then, the preliminary results of this study may be better suited for dissemination through a specialty journal.

Specific points:

1. The authors need to show levels of endogenous Scap and Insigs in all of their studies. As they know, the regulation of SREBP processing by cholesterol is intimately linked to the ratios of Scap/Insig/SREBP. Overexpression of one, but not the other, components of this pathway rarely leads to regulated SREBP processing. Does SPRING knockout lower levels of endogenous Scap (in addition to mislocalization), leading to lower precursor SREBPs and the mature product? Or does it affect levels of Insigs, which would also affect regulated SREBP processing? The results of the Espenshade group suggest that Scap recycling and Scap stability are linked.
2. A key point of the mechanistic studies is the interaction of SPRING with Scap which is shown through the studies of Figure 6. The IP studies in 6a need several important controls – according to the model, Scap should interact with SPRING, but not Insig, in the sterol-depleted case, and Scap should interact with Insig, but not SPRING, in the sterol-rich case. It would be nice to know whether the transmembrane domain of Scap is sufficient to interact with SPRING. The microscopy studies in 6b need co-staining with ER and Golgi markers to establish the localization of the eGFP-Scap.
3. PNGase is a useful control for glycosylation, but EndoH should be used to establish that the bands they see are matured glycans. Is the single identified N-linked glycan site responsible for all the bands - they have at least 4 glycosylated bands (Fig. S6)
4. Please comment on why the screens did not identify Insig1 or Insig2. Also, please use the more familiar names of Site-1 protease and Site-2 protease instead of MBTPS1 and MBTPS2.
5. Figure 2D needs Western blot results to verify SPRING knockout efficiency. For the mice in vivo knock-down experiment, it would be nice to know whether there were changes in hepatic lipids (especially cholesterol), blood LDL, and FFA levels.
6. Please provide details of how sterol depletion was carried out in all experiments. Treatment with 3 mM cyclodextrin in LPDS (Fig. 6E) seems excessive, cell viability should be shown.

Reviewer #2 (Remarks to the Author):

In 1990s, Brown and Goldstein's lab identified SREBP as a transcription factor controlling cholesterol homeostasis. In the ensuing decade, they identified a complex network of proteins regulating SREBP activation in a cholesterol-dependent manner. In particular, SCAP, a cholesterol sensor, escorts SREBP from the ER to the Golgi, where SREBP is cleaved to its active form by S1P and S2P proteases. It seemed that the entire SREBP pathway has already been worked out. Thus, it is a bit surprising that there were still uncovered regulators in the pathway. In this manuscript, using haploid genetic screens, the authors identified SPRING as a positive regulator in the SREBP pathway. The authors went on to demonstrate that SPRING regulates the proper localization of SCAP. Overall, this is an interesting finding. I have a few suggestions that may help the authors improve the manuscript.

1. In Figure 6, the authors showed that SCAP overexpression restored SREBP signaling in SPRING KO cells. This is an interesting observation but the authors need to also check whether SPRING KO decreases SCAP expression.
2. The authors need to examine SQLE expression in primary cells since SQLE is used as a readout for SPRING activity in these cells.
3. The authors need to examine the subcellular localization and expression levels of S1P and S2P in SPRING KO cells.
4. The expression levels of SPRING in HAP1 cells, mouse hepatocytes, HeLa cells etc need to be shown to confirm SPRING KO.

Reviewer #3 (Remarks to the Author):

In this manuscript, Loregger and colleagues report the identification of SPRING (C12ORF49) as a new positive regulator of the SREBP pathway, downstream gene expression and lipid homeostasis. Using multiple genetic screens in mammalian cells, the authors identify SPRING and demonstrate that it is required for SREBP activity in several mammalian cell lines and mouse primary hepatocytes. Mechanistic studies suggest that SPRING functions to recycle SCAP back to the ER from the Golgi following SREBP cleavage. The manuscript is well written, data are clearly presented, and overall the data support the authors' conclusions. Once fully supported, these conclusions will be very significant and of interest to broad audience: those interested in metabolic disease, lipid homeostasis and the cell biology of Golgi-to-ER transport. Reports of high impact discoveries should generate more questions than they answer. However, important questions remain to be address in this initial characterization of SPRING that are detailed below.

Major comments:

1. A major unaddressed question in the manuscript is whether SPRING function is specific/restricted to the SREBP pathway or whether it performs a more general role in Golgi-to-ER transport that affects SCAP and other proteins that have not been examined in the study. This is important as a protein-protein interaction was reported for overexpressed SCAP and SPRING, but not the endogenous proteins.

To help reviewers and readers better evaluate this issue, the authors should include complete datasets for their genetic screens as well at the RNAseq experiment in Fig 3A. For example, in Fig 1e, what is the gene below MBTPS1 and to the right of SCAP? Is that a COPI subunit? Might SPRING recruit COPI to the membrane. Was "general" secretory machinery identified in screens?

Further, in the WT vs SPRING KO RNA Seq experiment, many other genes are down-regulated more than SREBP pathway genes. What does GO term analysis reveal for these genes and these data overall? Are stress response pathways activated?

2. Data indicate that SPRING is required to recycle SCAP to ER. Localization of two ER proteins known to recycle from the Golgi-to-ER via COPI vesicles (for example some v-SNAREs) should be examined in SPRING KO cells to test whether SPRING function is specific to the SREBP pathway.

3. Data indicate that SPRING is required for SREBP pathway activity, but no experiments are presented to indicate that SPRING function or protein level is regulated. The authors should therefore refrain from using the terms "regulator" or "SPRING regulates" as in the running title.

4. The conclusion that SPRING is a Golgi resident protein rests on studies of overexpressed, tagged protein. One would expect that the N-linked sugars on SPRING would be EndoH-resistant. This experiment would provide further evidence that overexpressed protein is in the Golgi.

5. The KDEL tagging experiments are complicated to interpret given that the KDEL signal functions on soluble, luminal proteins and SPRING appears to be an integral membrane protein. Further, the topology of the N- and C-termini is unknown. Protease protection experiments should be performed to test whether the C-terminus SPRING is luminal.

6. The fact that SREBP1 target genes seem less affected by SPRING KD in the liver is very interesting. This point should be highlighted by including classic SREBP1c targets in Fig. 4C, such as FASN and SCD1.

6. In Fig. 2F, SQLE is still sterol-regulated when SREBP-N is overexpressed presumably due to sterol regulation of SQLE degradation. mRNA expression for SQLE or another SREBP target gene should be included as one would expect that mRNA levels are not regulated when only the N-terminus is expressed.

7. What is the expression pattern of SPRING in mouse tissues? Is it ubiquitously expressed or restricted to the liver?

Minor comments:

1. It is unclear as written whether SPRING homologs exist. Page 17, states "with no apparent homology to other proteins" but on page 19 in the Discussion, "belongs to an uncharacterized protein family (Pfam UPF0454)". Please clarify.

2. SREBP stands for sterol regulatory element-binding protein. Please review in manuscript as it is sometimes referred to as sterol-responsive and sterol response element.

3. p. 3 last sentence, the term "positive feedback" better describes the regulation compared to "feed-forward".

4. In Fig. 6A, please note the loading of total extract to bound, such that the fraction of SPRING bound can be determined. Please reduce cropping on SPRING blot so that it can be determined whether SCAP binds all forms of SPRING.

5. In Fig. 4C, please further describe how the y-axis "relative expression" was calculated. Are data normalized to control shRNA?

We would like to thank the reviewers for their constructive and positive evaluation of our study. In the revised manuscript we include a comprehensive set of new experiments and expand the discussion on the mechanistic implications of our study. Please find below a point-by-point response to the comments that were raised.

Reviewer #1

“In this study, the authors use a clever screening strategy using three independent approaches to uncover SPRING as a new factor involved in SREBP signaling and cholesterol metabolism. The screen identified known modulators of SREBPs – Scap, Site-1 protease, Site-2 protease – and a new gene, SPRING. Any new modulator of the SREBP pathway would be a major advance, therefore the finding of SPRING could be a big step forward. Unfortunately, the effects of knockdown or selective deletion of SPRING on SREBP signaling are quite modest (Figure 2D). Moreover, studies to elucidate the mechanism by which SPRING affects SREBP are not convincing. The authors claim that SPRING is a Golgi-resident protein that facilitates the recycling of Scap from the Golgi to the ER so that Scap can carry out another round of SREBP transport. The experiment where ER-localized SPRING-KDEL lowers levels of squalene epoxidase (a proxy for transport of Scap and SREBP) is clever. However, many of the authors findings could be attributed to lower levels of Scap, so their hypothesis is intriguing but not proven. Much more cell biology and biochemistry would need to be done to establish their mechanistic model. Until then, the preliminary results of this study may be better suited for dissemination through a specialty journal.”

We thank the reviewer for recognizing the novelty and potential impact of our finding to the field. In the revised manuscript a large collection of new experiments has been added to address his/her comments. Extending the role of SPRING in governing lipid metabolism, we include a bioinformatic analysis revealing that a wide-range of human tumors are strongly dependent for their growth on SPRING expression as part of an SREBP-centered node (new Supplementary Figure VIII). One of the findings highlighted in the revised manuscript is, as already suggested by this reviewer, that SCAP levels are reduced and SCAP is mislocalized in cells lacking SPRING.

1. The authors need to show levels of endogenous Scap and Insigs in all of their studies. As they know, the regulation of SREBP processing by cholesterol is intimately linked to the ratios of Scap/Insig/SREBP. Overexpression of one, but not the other, components of this pathway rarely leads to regulated SREBP processing. Does SPRING knockout lower levels of endogenous Scap (in addition to mislocalization), leading to lower precursor SREBPs and the mature product? Or does it affect levels of Insigs, which would also affect regulated SREBP processing? The results of the Espenshade group suggest that Scap recycling and Scap stability are linked.

The reviewer is correct in pointing out that SCAP recycling and stability are intimately linked, as demonstrated by the Espenshade group, and that altered SCAP or INSIG levels may contribute to the phenotype associated with loss of *SPRING*. For this reason, as requested, we have determined the levels of endogenous SCAP and INSIG1. As expected from loss of SREBP signaling in cells lacking *SPRING* we found that loss of *SPRING* resulted in an almost complete absence of endogenous INSIG1 protein under basal and sterol-depleted conditions (new Figure 2B). We also determined the level of SCAP in cells lacking *SPRING*. Consistent with *SPRING*'s potential role in SCAP trafficking we observed that endogenous SCAP levels are markedly reduced in Hap1. (new Figure 6C and new Supplementary Figures II-D). We have also determined SCAP levels in Hepa1-6 cells and observe a slight reduction (quantified ~20%) (Figure 1 for reviewer). It is conceivable that the more limited decrease in SCAP levels in this cell model is due to the fact that - different from the Hap1 cells - here we only used temporal ablation of *Spring* (4 days). Given this magnitude of change we are hesitant to include this result as such in the manuscript. The new results are discussed and included in the revised manuscript.

Note: We would like to point out that in the original submission (Figure 2F) we demonstrated that absence of *SPRING* reduces the level of precursor and mature SREBP2.

“2. A key point of the mechanistic studies is the interaction of SPRING with Scap which is shown through the studies of Figure 6. The IP studies in 6a need several important controls – according to the model, Scap should interact with SPRING, but not Insig, in the sterol-depleted case, and Scap should interact

with Insig, but not SPRING, in the sterol-rich case. It would be nice to know whether the transmembrane domain of Scap is sufficient to interact with SPRING. The microscopy studies in 6b need co-staining with ER and Golgi markers to establish the localization of the eGFP-Scap.”

We agree that the interaction between SCAP and INSIG or SPRING should be binary and dependent on the cellular sterol status. However, all three proteins are complex membrane proteins and their extraction requires stringent lysis conditions and detergents that severely limit the ability to study the relevant cellular compartment in which these interactions take place under physiological conditions, particularly as these experiments require over-expression of SCAP, INSIG, and SPRING. Moreover, we point out that at any given time, a fraction of SCAP will be found in both compartments irrespectively of the cellular sterol status due to its production and recycling, confounding the interpretation of this experiment. We did test whether SPRING and INSIG1 interact in co-IP experiments and found no interaction (not shown). Since this is a negative result we opted not to include it in the manuscript.

As requested, we include new images clearly delineating the mislocalization of eGFP-SCAP to the Golgi irrespectively of the cellular sterol status in CHO-eGFP-SCAP cells lacking *SPRING* (new Supplementary Figure VII-A). We agree that determining the functional domains in SCAP and SPRING that mediate their interaction is of interest, but respectfully submit that this goes beyond the immediate scope of the current study and is something we plan to address in the future.

“3. PNGase is a useful control for glycosylation, but EndoH should be used to establish that the bands they see are matured glycans. Is the single identified N-linked glycan site responsible for all the bands - they have at least 4 glycosylated bands (Fig. S6)”

To address the nature of the glycosylation on SPRING we performed de-glycosylation assays using EndoH, akin to the ones done with PNGase-F. The enzyme EndoH can remove glycans from ER- and *cis/medial* Golgi-resident proteins before they acquire complex glycans (*i.e.* post α -mannosidase II in the *medial* Golgi; *e.g.* PMID: 29725121). As shown in new Supplementary Figure VI-D, glycosylation on SPRING is removed by EndoH, indicating that it does not acquire complex glycans. Taken together with the observed localization of SPRING in the Golgi (Figure 5B), sensitivity to EndoH implies that SPRING is located in the *cis/medial* Golgi, which is coincidentally where the SCAP/SREBP/S1P machinery is located (PMID: 10500160, 10619424). Additionally, we have also generated a glycosylation-defective mutant in which we mutated the sole predicted glycosylation site (N67Q). Expression of this mutant resulted in SPRING protein lacking glycosylation supporting the prediction that this is the single glycosylation site in SPRING (new Figure 5C). Together our findings support localization of SPRING in the *cis/medial* Golgi. We discuss this in the revised manuscript.

“4. Please comment on why the screens did not identify Insig1 or Insig2. Also, please use the more familiar names of Site-1 protease and Site-2 protease instead of MBTPS1 and MBTPS2.”

Our SQLE screen actually did identify “hits” in INSIG1 and INSIG2. The number of gene-trap insertions that were identified was low (~50). The mutational index of INSIG2 did not reach statistical significance and hence it seems that in Hap1 cells under the tested conditions its role is limited. The mutational index of INSIG1, on the other hand, was significant and in line with its known function was found as a negative regulator of the SREBP pathway. In Figure 1E we have now highlighted INSIG1 and mention this in the results section.

We too regularly refer to Site-1 and -2 proteases when discussing the proteins. However, as throughout the manuscript we use the official gene names *MBTPS1* and *MBTPS2* when referring to these genes (*i.e.* Figure 1E,F,G), we think that it is correct to refer with similar nomenclature to the gene products. We add a clarification about this in the introduction section.

“5. Figure 2D needs Western blot results to verify SPRING knockout efficiency. For the mice in vivo knock-down experiment, it would be nice to know whether there were changes in hepatic lipids (especially cholesterol), blood LDL, and FFA levels.”

SPRING is a very lowly abundant protein. We have made great efforts to detect the endogenous SPRING protein in human and murine samples. Unfortunately, despite our efforts we were unable to detect endogenous SPRING in primary mouse hepatocytes, likely a result of low abundance and the typically limited protein content of these samples. However, we were able to successfully determine

endogenous SPRING levels in human Hap1 cells and in murine Hepa1-6 cells, and to show that the protein seems absent in the KO samples (new Figure 2B and Supplementary Figure IV-C).

In the original manuscript we actually indicated in the results section that *in vivo* we did not observe changes in hepatic lipids (p.17), despite a clear effect on the SREBP pathway. Similarly, we found no marked changes in plasma lipids under this experimental setup. As we indicated, this most likely reflects residual SPRING expression/activity, and the short-term nature of the experiment. Future experiments using more advanced *in vivo* SPRING models (e.g. conditional KO mice) are required to address the role of SPRING in regulating hepatic and plasma lipid levels. We clarify this point in the revised manuscript.

“6. Please provide details of how sterol depletion was carried out in all experiments. Treatment with 3 mM cyclodextrin in LPDS (Fig. 6E) seems excessive, cell viability should be shown.”

We have clarified the “Materials and Methods” section as requested. Briefly, our standard procedure is to culture cells in sterol-depletion medium (DMEM, IMDM, or DMEM/F12) supplemented with 10% lipoprotein-deficient serum (LPDS), 2.5 µg/mL simvastatin, and 100 µM mevalonate, as we and others in the field regularly do (e.g. PMID: 28882874). For specifically evaluating cholesterol synthesis (experiment in Figure 6F) we used a different regimen that maintains HMGCR activity in which we culture the cells in medium containing 10% LPDS and 3 mM β-methyl-cyclodextrin. We point out that the sensitivity to β-methyl-cyclodextrin is cell-line/type dependent and the choice of using 3mM was made after assessing cell viability. As requested, we provide new Supplementary Figure VII-B to demonstrate that under the same experimental conditions used for the assay cell viability was not affected.

Reviewer #2

“In 1990s, Brown and Goldstein’s lab identified SREBP as a transcription factor controlling cholesterol homeostasis. In the ensuing decade, they identified a complex network of proteins regulating SREBP activation in a cholesterol-dependent manner. In particular, SCAP, a cholesterol sensor, escorts SREBP from the ER to the Golgi, where SREBP is cleaved to its active form by S1P and S2P proteases. It seemed that the entire SREBP pathway has already been worked out. Thus, it is a bit surprising that there were still uncovered regulators in the pathway. In this manuscript, using haploid genetic screens, the authors identified SPRING as a positive regulator in the SREBP pathway. The authors went on to demonstrate that SPRING regulates the proper localization of SCAP. Overall, this is an interesting finding. I have a few suggestions that may help the authors improve the manuscript.”

We thank the reviewer for his comments and suggestions on how to improve our manuscript. Like the reviewer, we too were surprised and excited that we were able to find a new player in the core SREBP machinery and to contribute to the groundbreaking model developed by the Brown and Goldstein lab. Extending the role of SPRING in governing lipid metabolism, we include a bioinformatic analysis revealing that a wide-range of human tumors are strongly dependent on SPRING expression as part of an SREBP-centered node.

“1. In Figure 6, the authors showed that SCAP overexpression restored SREBP signaling in SPRING KO cells. This is an interesting observation but the authors need to also check whether SPRING KO decreases SCAP expression.”

The reviewer brings up a good point, particularly as the Espenshade group has recently demonstrated that SCAP recycling and stability are intimately linked. For this reason, as requested, we have determined the level of endogenous SCAP. Consistent with SPRING’s potential role in SCAP recycling, we observed that endogenous SCAP levels are markedly reduced in Hap1 cells lacking *SPRING* (new Figure 6C and new Supplementary Figures II-D). As a result, these cells lack functional SCAP to support SREBP maturation and hence can be rescued, as we show, by reintroducing heterologous SCAP. We have also determined SCAP levels in Hepa1-6 cells and observe a slight reduction (quantified ~20%) (Figure 1 for reviewer). It is conceivable that the more limited decrease in SCAP levels in this cell model is due to the fact that - different from the Hap1 cells - here we only use temporal ablation of *Spring* (4 days). Given this magnitude of change we are hesitant to include this result as such in the manuscript. These new results are discussed and included in the revised manuscript.

“2. The authors need to examine SQLE expression in primary cells since SQLE is used as a readout for SPRING activity in these cells.”

This is a good suggestion. The Zelcer lab has a long-standing interest in the post-transcriptional regulation of SQLE and has an active line of work on this subject. We regularly and readily detect endogenous human SQLE (e.g. Figure 1B). However, detecting endogenous mouse SQLE has not been successful despite testing at least 6 commercial anti-SQLE antibodies, and 2 rabbit polyclonal antibodies against mouse SQLE that we generated ourselves. As such, we unfortunately can't include SQLE detection in this experiment, but instead show other markers that faithfully report on the status of the SREBP pathway.

“3. The authors need to examine the subcellular localization and expression levels of S1P and S2P in SPRING KO cells.”

We tested the level of S1P in cells lacking SPRING and found that it was not different from control cells (new Supplementary Figure II-B). Localization of S1P in these cells also seemed unaltered (Figure 2 for reviewer). As we were unable to obtain reliable S2P-focused reagents we opted to address the intactness of the S1P/S2P axis by following the processing of the ER-stress-related factor ATF6 that undergoes processing by these proteases in a manner akin to that of SREBPs. In new Supplementary Figure II-C we show that basal ER-stress as evaluated by expression of ATF6-driven genes, and other ER-related genes, is not changed in cells lacking SPRING. Importantly, induction of ER stress by Tunicamycin, which promotes ATF6 translocation to the Golgi and subsequent proteolytic processing by S1P/S2P proteases was intact, albeit we did observe a small yet significant reduction that we discuss in the “Discussion” section in relation to the phenotype of cells lacking SPRING. At the very least this indicates that the S1P/S2P axis is not completely abrogated in cells lacking SPRING and can respond to physiological cues. Moreover, as we also pointed out in the original submission, processing of SREBP and downstream gene activation (c.f. Figure 2A, 2E) is not completely abolished but is rather severely attenuated.

“4. The expression levels of SPRING in HAP1 cells, mouse hepatocytes, HeLa cells etc need to be shown to confirm SPRING KO.”

For all the models used in the study we provide mRNA expression data for *SPRING*. We have observed that *SPRING* is a very lowly abundant protein. We were able to successfully determine endogenous *SPRING* levels in human Hap1 cells and in murine Hepa1-6 hepatocytes, and to show that the protein is absent in the KO samples (new Figure 2B and new Supplementary Figure IV-C). We point out that we have not made *SPRING* KO HeLa cells in our study, but only used them to localize *SPRING* to the Golgi.

Reviewer #3

“In this manuscript, Loriger and colleagues report the identification of SPRING (C12ORF49) as a new positive regulator of the SREBP pathway, downstream gene expression and lipid homeostasis. Using multiple genetic screens in mammalian cells, the authors identify SPRING and demonstrate that it is required for SREBP activity in several mammalian cell lines and mouse primary hepatocytes. Mechanistic studies suggest that SPRING functions to recycle SCAP back to the ER from the Golgi following SREBP cleavage. The manuscript is well written, data are clearly presented, and overall the data support the authors' conclusions. Once fully supported, these conclusions will be very significant and of interest to broad audience: those interested in metabolic disease, lipid homeostasis and the cell biology of Golgi-to-ER transport. Reports of high impact discoveries should generate more questions than they answer. However, important questions remain to be address in this initial characterization of SPRING that are detailed below.”

We are grateful to the reviewer for his positive evaluation of our study, its potential impact to various fields, and the new questions it raises. This notion is further strengthened by inclusion of a bioinformatic analysis revealing that a wide-range of human tumors are strongly dependent on *SPRING* expression as part of an SREBP-centered node.

“Major comments:

1. A major unaddressed question in the manuscript is whether SPRING function is specific/restricted to the SREBP pathway or whether it performs a more general role in Golgi-to-ER transport that affects SCAP and other proteins that have not been examined in the study. This is important as a protein-protein interaction was reported for overexpressed SCAP and SPRING, but not the endogenous proteins.

To help reviewers and readers better evaluate this issue, the authors should include complete datasets for their genetic screens as well as the RNAseq experiment in Fig 3A. For example, in Fig 1e, what is the gene below MBTPS1 and to the right of SCAP? Is that a COPI subunit? Might SPRING recruit COPI to the membrane. Was "general" secretory machinery identified in screens? Further, in the WT vs SPRING KO RNA Seq experiment, many other genes are down-regulated more than SREBP pathway genes. What does GO term analysis reveal for these genes and these data overall? Are stress response pathways activated?"

The reviewer brings up the important question of how specific is SPRING for the SREBP pathway. We can't formally rule out that SPRING activity is limited to the SREBP pathway only, yet focused on this as our independent genetic screens pointed towards SPRING's involvement in this pathway. This notion is also further supported by the co-dependency between SPRING, SCAP, SREBP, MBTPS1, and MBTPS2, as seen in a wide range of human tumors (new Supplementary Figures VIII-A,B). Moreover, the Brummelkamp group has conducted >100 haploid genetic screens reporting on a wide range of cellular processes, and amongst these *SPRING* was identified as a hit in the lipid-focused ones (Figure 3 for reviewer). As requested, we also conducted a GO analysis to identify cellular pathways that are differentially represented in control vs *SPRING* KO cells (new Supplementary Figure III-B). While this analysis identified several statistically significantly altered pathways, none directly related to the secretory machinery, COPI-mediated transport, or stress were identified. The relevance of these pathways to the cellular phenotype is not evident, and it is possible that these represent a secondary adaptation.

In relation to ER stress, we specifically evaluated whether basal and tunicamycin-induced ER stress is different in the absence of *SPRING*. In new Supplementary Figure II-C we show that basal ER-stress as evaluated by expression of ER stress-related genes is not changed in cells lacking *SPRING*. Importantly, induction of ER stress by Tunicamycin that requires ATF6 proteolytic processing by S1P and S2P -similar to SREBP - was intact, albeit we did observe a small yet significant reduction that we discuss in the "Discussion" section in relation to the phenotype of cells lacking *SPRING*.

The gene "below MBTPS1 and to the right of SCAP" the reviewer inquires about is CREBBP, and not a COPI subunit. We have now indicated this in Figure 1E. All the datasets will be made available through public repositories upon publication. We did not follow up on CREBBP, but one could speculate that this may be related to its ability to acetylate and stabilize/activate SREBP (e.g. PMID: 12640139, 8918891).

"2. Data indicate that SPRING is required to recycle SCAP to ER. Localization of two ER proteins known to recycle from the Golgi-to-ER via COPI vesicles (for example some v-SNAREs) should be examined in SPRING KO cells to test whether SPRING function is specific to the SREBP pathway."

We appreciate the reviewer's interest in studying COPI-mediated vesicular transport in the context of *SPRING*. However, as detailed in our response to question #1 above, we do not see global alterations consistent with defective COPI-mediated vesicular transport, nor do our screens identify any obvious COPI-related components. We therefore hope that the reviewer can appreciate that studying COPI-mediated transport in relation to *SPRING* was not our direct priority and represents a comprehensive project on its own. Accordingly, we adapted the "Discussion" to adequately clarify our view on the potential mechanism underlying *SPRING* function.

"3. Data indicate that SPRING is required for SREBP pathway activity, but no experiments are presented to indicate that SPRING function or protein level is regulated. The authors should therefore refrain from using the terms "regulator" or "SPRING regulates" as in the running title."

We have changed the running title and scanned the manuscript to conform with this comment. We point out that we did not make any comment or claim regarding *SPRING* itself or its function being regulated in our manuscript. In fact, in Figure 2A we show that the mRNA level of *SPRING* is not sensitive to the

cellular sterol status. Additionally, in the revised manuscript we now demonstrate that the level of SPRING protein itself is also not sterol-responsive (new Figure 2B).

“4. The conclusion that SPRING is a Golgi resident protein rests on studies of overexpressed, tagged protein. One would expect that the N-linked sugars on SPRING would be EndoH-resistant. This experiment would provide further evidence that overexpressed protein is in the Golgi.”

To address the nature of the glycosylation on SPRING we performed de-glycosylation assays using EndoH, akin to the ones done with PNGase-F. The enzyme EndoH removes glycans from ER- and *cis/medial* Golgi-resident proteins before they acquire complex glycans (*i.e.* post α -mannosidase II in the *medial* Golgi; *e.g.* PMID: 29725121). As shown in new Supplementary Figure VI-D, glycosylation on SPRING is removed by EndoH, indicating that it does not acquire complex glycans. Taken together with the observed localization of SPRING in the Golgi (Figure 5B), sensitivity to EndoH implies that SPRING is located in the *cis/medial* Golgi, which is coincidentally where the SCAP/SREBP/S1P machinery is located (PMID: 10500160, 10619424). Additionally, we have also generated a glycosylation-defective mutant in which we mutated the sole predicted glycosylation site (N67Q). Expression of this mutant resulted in SPRING protein lacking glycosylation supporting the prediction that this is the single glycosylation site in SPRING (new Figure 5C). Together our findings support localization of SPRING in the *cis/medial* Golgi. We discuss this in the revised manuscript.

“5. The KDEL tagging experiments are complicated to interpret given that the KDEL signal functions on soluble, luminal proteins and SPRING appears to be an integral membrane protein. Further, the topology of the N- and C-termini is unknown. Protease protection experiments should be performed to test whether the C-terminus SPRING is luminal.”

Following the suggestion of the reviewer we conducted protease protection assays to assess the topology of the N- and C-termini of SPRING. As shown in new Figure 5D, SPRING's C-terminus was protected from trypsin proteolytic activity in the absence of a membrane-permeabilizing detergent. This is in strong contrast to our parallel experiment with a C-terminally HA-tagged LDLR construct where it is known that the C-terminus faces the cytoplasm. These results support that contention that SPRING's C-terminal faces the luminal side of the Golgi, or the ER in the case of the KDEL mutant. These results are included and discussed in the revised manuscript.

“6. The fact that SREBP1 target genes seem less affected by SPRING KD in the liver is very interesting. This point should be highlighted by including classic SREBP1c targets in Fig. 4C, such as FASN and SCD1.”

We agree with the reviewer that the lack of change in the SREBP1 pathway is intriguing and will require further examination. As requested, we have included expression of *Fasn*, *Scd1*, and *Acc* in revised Figure 4C.

6. In Fig. 2F, SQLE is still sterol-regulated when SREBP-N is overexpressed presumably due to sterol regulation of SQLE degradation. mRNA expression for SQLE or another SREBP target gene should be included as one would expect that mRNA levels are not regulated when only the N-terminus is expressed.

The reviewer is correct regarding his comment on the sterol-regulated degradation of SQLE, as we and the Brown lab have reported (*e.g.* PMIDs: 26527619, 24449766, 21356516). As for the N-terminal SREBP domain used in this study; this domain represents the mature (*i.e.* transcriptionally active) SREBP2 transcription factor. This domain, initially identified by the Brown and Goldstein lab (*e.g.* PMID: 7903453, 9062341), has been shown by multiple groups worldwide to be sufficient to drive expression of SREBP-dependent transcription. It was used in the experiment in Figure 2G to directly demonstrate that SPRING KO cells do not have an intrinsic lesion in transcriptional activation of the SREBP pathway, since when these cells express the constitutively active SREBP transcription factor they can support SREBP signaling. To demonstrate this point again we include a figure for the reviewer in which we introduced the same construct used in the figure into cells and show induction of SREBP target genes and corresponding proteins (Figure 4 for reviewer).

“7. What is the expression pattern of SPRING in mouse tissues? Is it ubiquitously expressed or restricted to the liver?”

We have determined the mRNA expression of *Spring* in a panel of mouse tissues. Expression is ubiquitous, albeit somewhat higher in the liver and kidney, consistent with a general role in regulating the widely conserved SREBP pathway. This is now included as new Supplementary Figure V-A and discussed in the results section.

“Minor comments:

1. It is unclear as written whether SPRING homologs exist. Page 17, states "with no apparent homology to other proteins" but on page 19 in the Discussion, "belongs to an uncharacterized protein family (Pfam UPF0454). Please clarify.”

SPRING and its orthologs represent the members of this uncharacterized protein family. We clarify this in the text.

“2. SREBP stands for sterol regulatory element-binding protein. Please review in manuscript as it is sometimes referred to as sterol-responsive and sterol response element.”

Thanks for catching and pointing this out. We have changed the manuscript accordingly.

“3. p. 3 last sentence, the term "positive feedback" better describes the regulation compared to "feed-forward”.

Corrected.

“4. In Fig. 6A, please note the loading of total extract to bound, such that the fraction of SPRING bound can be determined. Please reduce cropping on SPRING blot so that it can be determined whether SCAP binds all forms of SPRING.”

We have adjusted the figure and point out that the fraction of SPRING that binds to SCAP is small, yet the results were very reproducible. We mention this in the results section.

5. In Fig. 4C, please further describe how the y-axis "relative expression" was calculated. Are data normalized to control shRNA?

The Y-axis represents the mRNA levels normalized to the reference gene 36B4 and not to the control shRNA. We corrected the Y-axis to reflect this.

Review figure 1

Figure 1. Hepa1-6-Cas9 cells were infected with Ad-3xgRNA-mSpring or Ad-3xgRNA-GFP at an MOI of 50 for 96 hours. Subsequently, membranes were isolated and immunoblotted as indicated against mSCAP or GM130 (loading control). Results from 3 independent experiments are shown.

Note that mSCAP was detected using a different antibody (C-20) than that used for detection of human SCAP.

Review figure 2

Figure 2. Hap1-wildtype and Hap1-SPRING^{KO} cells were plated on coverslips and 24 hrs later immunostained as indicated. Nuclei were counterstained with DAPI.

Review figure 3

Figure 3. The mutational index (MI) of *SPRING* is plotted across all individual screens conducted in the Brummelkamp lab. Note that *SPRING* is identified in lipid-related screens (highlighted in blue on the right).

Review figure 4

Figure 3. HEK293T cells were transfected with a control (empty vector) or an expression plasmid encoding the N-terminal domain of SREBP2 which results in production of constitutively active SREBP2 protein (i.e. there is no need for proteolytic processing). 48 hrs post-transfection cells were harvested for (left) analysis of mRNA levels by qPCR, or (right) analysis of protein levels by immunoblotting as indicated for both. Results from three independent experiments are shown.

Reviewers' comments:

Reviewer #1 (Remarks to the Author):

In this revised version, the authors have addressed all the points that I raised. They don't necessarily provide conclusive answers to all the questions regarding the mechanisms of how SPRING works, but they do provide an extensive and conscientious revision. So, even though the mechanistic questions are not fully resolved, the observation of SPRING as a potential modulator of the SREBP pathway will be of great interest to the field and will spur future work.

Reviewer #2 (Remarks to the Author):

I reviewed the revisions related to my previous comments. I found the authors have satisfactorily addressed my concerns. For experiments that are infeasible at the moment, the authors designed alternative strategies to answer the question. For example, ER stress-induced ATF6 activation was used to assess S2P activity, which was a clever strategy.

Reviewer #3 (Remarks to the Author):

The authors have addressed the majority of my comments regarding the initial submission. Using both in vitro and in vivo models, they convincingly show that SPRING is required for proper function of the SREBP pathway through control of SCAP function. In doing so, they define a new determinant of SREBP pathway function and this information will be of broad interest.

However, key questions from the previous review remain unanswered:

1. Is SPRING specific for this pathway?

To test this, the authors should examine the localization of other cycling secretory pathway proteins (previous Major Comment #2).

2. Does SPRING directly control SCAP function as suggested by their model figure?

Figure 6A is the only experiment addressing whether SPRING functions directly on SCAP.

Interestingly, SCAP appears to bind preferentially to the ER glycosylated form of SPRING, which is unexpected based on the authors' proposed model. To rule out that the SCAP-SPRING interaction results from incomplete solubilization of ER membranes, the authors should examine binding of SCAP to another ER protein such as SQLE or SQS in the same pull down.

Additional comments:

1. Consistent with previous Major Comment #3, please consider revising the second sentence of the Abstract: "Using a suite of human haploid genetic screens, we identify the SREBP Regulating Gene (SPRING; C12ORF49) as a novel regulator this pathway." Similar comment for heading for Figure 6.

2. Regarding Minor Comment #2, the Abbreviations still refer to SRE as "sterol response element".

3. Regarding Minor Comment #4, I am still unable to determine the percentage/fraction of SPRING that co-purifies with SCAP. Please indicate in the legend whether the bound fraction is enriched relative to the total lysate and to what extent.

Reviewer #1

“In this revised version, the authors have addressed all the points that I raised. They don't necessarily provide conclusive answers to all the questions regarding the mechanisms of how SPRING works, but they do provide an extensive and conscientious revision. So, even though the mechanistic questions are not fully resolved, the observation of SPRING as a potential modulator of the SREBP pathway will be of great interest to the field and will spur future work.”

We thank the reviewer for his/her positive evaluation and for recognizing our effort and the importance of our study.

Reviewer #2

“I reviewed the revisions related to my previous comments. I found the authors have satisfactorily addressed my concerns. For experiments that are infeasible at the moment, the authors designed alternative strategies to answer the question. For example, ER stress-induced ATF6 activation was used to assess S2P activity, which was a clever strategy.”

We thank the reviewer for his/her positive evaluation and for recognizing our effort and the importance of our study.

Reviewer #3

“The authors have addressed the majority of my comments regarding the initial submission. Using both in vitro and in vivo models, they convincingly show that SPRING is required for proper function of the SREBP pathway through control of SCAP function. In doing so, they define a new determinant of SREBP pathway function and this information will be of broad interest.”

We thank the reviewer for his/her positive assessment and for recognizing the importance of our study and its relevance to the field.

“However, key questions from the previous review remain unanswered:

1. Is SPRING specific for this pathway?

To test this, the authors should examine the localization of other cycling secretory pathway proteins (previous Major Comment #2).”

The primary goal of our study was to identify novel SREBP modifiers. With this in mind, we would like to respectfully point out that we were careful not to state or suggest that SPRING is exclusively specific for the SREBP pathway.

In our revised manuscript, not only do we not suggest that SPRING is specific for the SREBP pathway, but we in fact provide RNAseq data and GO pathway analysis identifying differentially expressed pathways in the absence of SPRING (Supplementary Figure III). However, none of these could be directly linked to the observed alterations in the SREBP pathway, or related to the secretory machinery, COPI-mediated transport, or general stress response as was asked. As we pointed out, the relevance of these altered pathways to the cellular phenotype was not evident, and it is possible that these represent a secondary adaptation.

More importantly, we provide experimental evidence that the ATF6-mediated ER-stress response elicited by tunicamycin is attenuated in the absence of SPRING (Supplementary Figure II). The two-step, sequential proteolytic activation of ATF6 by S1P/S2P resembles that of SREBPs. However, an important distinction is that ATF6 activation does not involve SCAP. This may suggest that SPRING plays a role in modulating S1P/S2P-mediated activation of these transcription factors, through a yet undefined manner.

Finally, we would also like to point out that we have refrained from claiming that SPRING is required for retrograde recycling of SCAP to the ER. We concede that the exact mechanism underlying SPRING's modulation of the SREBP pathway is not fully elucidated, and that at present we are considering several

potential mechanisms that are consistent with our findings, including regulation of S1P activity and control of SCAP recycling. While we look forward to addressing these possibilities in the future, we respectfully submit that this is beyond the immediate scope of our study and detracts from our key finding, the identification of a previously uncharacterized gene, *SPRING*, as a novel *in vitro* and *in vivo* modulator of the SREBP pathway, and that the lesion in SREBP signaling in the absence of *SPRING* is at the level of functional SCAP. Please note that in the “Discussion” section we extensively discuss these potential mechanisms. We have included an additional statement on *SPRING* specificity in this section and also expand on potential mechanisms in our “graphical abstract” figure to indicate that we are referring to functional SCAP deficiency.

2. Does SPRING directly control SCAP function as suggested by their model figure? Figure 6A is the only experiment addressing whether SPRING functions directly on SCAP. Interestingly, SCAP appears to bind preferentially to the ER glycosylated form of SPRING, which is unexpected based on the authors' proposed model. To rule out that the SCAP-SPRING interaction results from incomplete solubilization of ER membranes, the authors should examine binding of SCAP to another ER protein such as SQLE or SQS in the same pull down.

In the manuscript we provide several lines of evidence to support Golgi localization of *SPRING* (Figure 5). Importantly, in the same figure we demonstrate that Golgi localization is critical for modulating the SREBP pathway, as ER-retained *SPRING* is unable to rescue the SREBP-lesion in cells lacking *SPRING*. It is not completely clear to us why the reviewer deduces that SCAP preferentially binds the ER form of *SPRING*, as one cannot infer this directly from the glycosylation pattern. Moreover, as also Golgi-localized proteins travel through the ER, at any given time there will be a fraction of both proteins in the ER and Golgi that in combination with the use of a harsh detergent buffer (RIPA), which is required to extract these membrane proteins, further limits drawing conclusions on the cellular interaction site. As such, we cannot formally rule out that the initial interaction between SCAP and *SPRING* takes place in the ER, but emphasize again that for *SPRING* to exert its activity it needs to reach the Golgi.

As requested by the reviewer, in our pull-down experiments we also evaluated the interaction of SCAP with the prototypic ER-resident protein INSIG. We did not include this result as we did not consider it necessary for evaluation of our experiments. However, given the reviewer's request we now modified Figure 6A to include this result showing that under the same IP conditions SCAP interacts with INSIG. Importantly, we include Figure 1 for the reviewer showing that INSIG (ER) and *SPRING* (Golgi) do not interact under the same experimental conditions used for studying the interaction with SCAP, in line with the notion that the SCAP-*SPRING* interaction is specific and not due to incomplete solubilization of ER membranes. These points are mentioned in the “Results” section.

“Additional comments:

1. Consistent with previous Major Comment #3, please consider revising the second sentence of the Abstract: "Using a suite of human haploid genetic screens, we identify the SREBP Regulating Gene (SPRING; C12ORF49) as a novel regulator this pathway." Similar comment for heading for Figure 6.”

We have replaced “regulator” with “determinant” and “regulates” with “modulates” in the abstract and heading of Figure 6, respectively.

“2. Regarding Minor Comment #2, the Abbreviations still refer to SRE as "sterol response element.”

We had misunderstood the initial comment and corrected the abbreviation for SRE so it now indicates “sterol regulatory element”

“3. Regarding Minor Comment #4, I am still unable to determine the percentage/fraction of SPRING that co-purifies with SCAP. Please indicate in the legend whether the bound fraction is enriched relative to the total lysate and to what extent.”

Please note that we immunoprecipitated SCAP and that there is a clear enrichment of SCAP in the IP fraction. With regard to *SPRING*, we do not find enrichment compared to the total cell lysates. As we pointed out in our initial response this may be due to the harsh nature of the buffers required to extract these membrane proteins, or possibly due to a weak or temporal nature of this interaction. We are therefore cautious to comment on the strength of the interaction and rather interpret these results

qualitatively to indicate that under these experimental conditions we are able to co-IP SPRING with SCAP. We mention this in the figure legend as requested.

Figure 1 for reviewer. HEK293T cells were transfected as indicated. Subsequently, Insig1-Myc was IP (IP:Myc) and total cell lysates and IP fractions were immunoblotted as indicated.